# EROS is a selective chaperone regulating the phagocyte NADPH oxidase and purinergic signalling

Lyra O Randzavola[1†], Paige M Mortimer[1†], Emma Garside[1], Elizabeth R Dufficy[2], Andrea Schejtman[3], Georgia Roumelioti[4], Lu Yu[4], Mercedes Pardo[4], Kerstin Spirohn[5,6,7], Charlotte Tolley[8], Cordelia Brandt[8], Katherine Harcourt[8], Esme Nichols[1], Mike Nahorski[9], Geoff Woods[9], James C Williamson[2,10], Shreehari Suresh[2], John M Sowerby[2,10], Misaki Matsumoto[11], Celio XC Santos[12], Cher Shen Kiar[13], Subhankar Mukhopadhyay[13], William M Rae[2,10], Gordon J Dougan[2], John Grainger[4,14], Paul J Lehner[2,10], Michael A Calderwood[5,6,7], Jyoti Choudhary[4], Simon Clare[8], Anneliese Speak[8], Giorgia Santilli[3], Alex Bateman[15], Kenneth GC Smith[2,10], Francesca Magnani[16]*, David C Thomas[1]*

[1]Department of Immunology and Inflammation, Centre for Inflammatory Disease, Imperial College London, London, United Kingdom; [2]The Department of Medicine, University of Cambridge School of Clinical Medicine, Cambridge, United Kingdom; [3]Molecular Immunology Unit, UCL Great Ormond Street Institute of Child Health, London, United Kingdom; [4]Functional Proteomics, Division of Cancer Biology, Institute of Cancer Research, London, United Kingdom; [5]Center for Cancer Systems Biology (CCSB), Dana-Farber Cancer Institute, Boston, United States; [6]Department of Genetics, Blavatnik Institute, Harvard Medical School, Boston, United States; [7]Department of Cancer Biology, Dana-Farber Cancer Institute, Boston, United States; [8]Wellcome Trust Sanger Institute, Hinxton, United Kingdom; [9]Cambridge Institute of Medical Research, University of Cambridge, Cambridge, United Kingdom; [10]Cambridge Institute of Therapeutic Immunology & Infectious Disease, Jeffrey Cheah Biomedical Centre Cambridge Biomedical Campus, Cambridge, United Kingdom; [11]Department of Pharmacology, Kyoto Prefectural University of Medicine, Kyoto, Japan; [12]School of Cardiovascular Medicine and Sciences, James Black Centre, King's College London, London, United Kingdom; [13]Peter Gorer Department of Immunobiology, School of Immunology & Microbial Sciences, King's College London, London, United Kingdom; [14]Lydia Becker Institute of Immunology and Inflammation, Faculty of Biology, Medicine and Health, University of Manchester, Manchester, United Kingdom; [15]European Molecular Biology Laboratory, European Bioinformatics Institute, Wellcome Trust Genome Campus, Hinxton, United Kingdom; [16]Department of Biology and Biotechnology, University of Pavia, Pavia, Italy

*For correspondence:
francesca.magnani@unipv.it (FM);
david.thomas1@imperial.ac.uk (DCT)

[†]These authors contributed equally to this work

Competing interest: The authors declare that no competing interests exist.

**Abstract** EROS (essential for reactive oxygen species) protein is indispensable for expression of gp91*phox*, the catalytic core of the phagocyte NADPH oxidase. EROS deficiency in humans is a novel cause of the severe immunodeficiency, chronic granulomatous disease, but its mechanism of action was unknown until now. We elucidate the role of EROS, showing it acts at the earliest stages of gp91*phox* maturation. It binds the immature 58 kDa gp91*phox* directly, preventing gp91*phox* degradation and allowing glycosylation via the oligosaccharyltransferase machinery and the incorporation of the heme prosthetic groups essential for catalysis. EROS also regulates the purine

receptors P2X7 and P2X1 through direct interactions, and P2X7 is almost absent in EROS-deficient mouse and human primary cells. Accordingly, lack of murine EROS results in markedly abnormal P2X7 signalling, inflammasome activation, and T cell responses. The loss of both ROS and P2X7 signalling leads to resistance to influenza infection in mice. Our work identifies EROS as a highly selective chaperone for key proteins in innate and adaptive immunity and a rheostat for immunity to infection. It has profound implications for our understanding of immune physiology, ROS dysregulation, and possibly gene therapy.

## Editor's evaluation

This valuable study focus follows this group's previous work on EROS and NOX2. In this current study, the authors examine neutrophil EROS in the generation of superoxide by the NADPH oxidase. They convincingly demonstrate how EROS is involved in the maturation of gp91phox and expand our knowledge of the role of EROS in regulating the expression of the P2x7 ion channel. This work will be of interest to biochemists and immunologists.

## Introduction

The phagocyte nicotinamide adenine dinucleotide phosphate (NADPH oxidase) generates reactive oxygen species (ROS) for host defence and is a critical component of innate immunity. This multi-subunit protein complex consists of (i) a membrane-bound heterodimer, gp91*phox*-p22*phox,* and (ii) the cytosolic components p67*phox* (*Volpp et al., 1988*; *Nunoi et al., 1988*), p47*phox* (*Volpp et al., 1988*; *Segal et al., 1985*), p40*phox* (*Wientjes et al., 1993*), and either Rac1 (*Abo et al., 1991*) or Rac2 (*Knaus et al., 1991*). When activated by microbial stimuli, the complex facilitates the transfer of electrons from cytosolic NADPH through the gp91*phox* (Nox2) protein to molecular oxygen, located either extracellularly or within phagosomes (*Segal, 2005*; *Thomas, 2017a*), generating superoxide anions. Various chemical reactions then drive the production of further antimicrobial ROS such as hydrogen peroxide ($H_2O_2$) and hypochlorous acid. Moreover, oxidation of key cysteine residues driven by $H_2O_2$ has been implicated in regulating other immune pathways such as inflammasome activation (*Meissner et al., 2010*), type 1 interferon production (*Holmdahl et al., 2016*; *Sareila et al., 2017*; *Olsson et al., 2017*; *Zhong et al., 2018*), LC3-associated phagocytosis (*Martinez et al., 2015*; *Martinez et al., 2016*) and autophagy (*Thomas, 2018a*, *deLuca et al., 2014*).

The importance of the phagocyte NADPH oxidase is underlined by chronic granulomatous disease (CGD), a severe monogenic immunodeficiency caused by loss of individual components, which presents as susceptibility to infections with catalase-positive organisms (including *Staphylococcus aureus*, *Salmonella,* and *Burkholderia* species) but also with autoinflammatory manifestations, characterised by sterile granulomatous inflammation (*Alimchandani et al., 2013*; *Salvator et al., 2015*; *Goldblatt et al., 1999*). Conversely, excess ROS generation can be damaging to tissues via, for example, lipid peroxidation and have been implicated in the pathogenesis of autoimmunity (*Choi et al., 2015*; *Hartung et al., 1988*; *Noubade et al., 2014*). Polymorphisms in the genes encoding subunits are also implicated in numerous autoimmune diseases (*Magnani et al., 2014*).

One way to control ROS generation is to regulate the abundance of gp91*phox*-p22*phox*, the critical membrane-bound components that facilitate electron transfer from NADPH to oxygen. These two proteins depend on one another for stable expression (*Segal, 1987*). gp91*phox* is synthesised in the endoplasmic reticulum (ER) as a 58 kDa polypeptide. It becomes a 65 kDa high-mannose precursor, then acquires heme, forms a heterodimer with p22*phox,* and is glycosylated in the Golgi apparatus before transport to endosomes (in macrophages) and to peroxidase-negative granules (in neutrophils) or the plasma membrane. The process of heterodimer formation is relatively inefficient, and gp91*phox* monomers are rapidly degraded from the ER (*DeLeo et al., 2000*). Chaperones such as hsp90 (*Chen et al., 2011*) and hsp70 (*Chen et al., 2012*) regulate gp91*phox* abundance, as does the negative regulator of reactive oxygen species (NRROS) (*Noubade et al., 2014*). We demonstrated that the protein essential for reactive oxygen species (EROS, gene symbol *CYBC1*) has profound effects on gp91*phox* and p22*phox* abundance. The heterodimer is essentially absent in EROS-deficient cells in both mouse (*Thomas et al., 2017b*) and human (*Thomas et al., 2018b*) immune cells, leading

to extreme susceptibility to infection. Further, we and others have demonstrated that human EROS deficiency is a novel cause of CGD: OMIM 618935 (*Thomas et al., 2018b*, *Arnadottir et al., 2018*).

However, important questions remain, including (i) the exact mechanism by which EROS regulates gp91*phox*-p22*phox* abundance and (ii) whether EROS regulates the expression of other proteins. Both are key questions given that EROS deficiency is associated with severe morbidity and some clinical manifestations not seen in CGD, such as autoimmune haemolytic anaemia (*Thomas et al., 2018b*) and glomerulonephritis (*Arnadottir et al., 2018*).

In this study, we show that EROS co-transfection with gp91*phox* markedly increases gp91*phox* expression in a variety of cell lines, selectively enhancing expression of the immature 58 kDa form of the protein and preventing its degradation. EROS can exist in a trimeric complex with gp91*phox* and p22*phox* through direct binding to gp91*phox*. Heme is not required for the interaction of gp91*phox* and EROS. EROS localises at the ER and the perinuclear membrane, interacting with components of the oligosaccharyltransferase complex (OST), consistent with a role early in biosynthesis. We also report the ligand-gated ion channel P2X7 as an EROS target. EROS binds directly to and co-immunoprecipitates with P2X7 and other members of the P2X family. Accordingly, P2X7-driven calcium flux, inflammasome activation, and surface receptor shedding are markedly abnormal in EROS-deficient cells with profound effects on both macrophage and T cell physiology. Together, the lack of gp91*phox*-p22*phox* and P2X7 leads to resistance to influenza A infection.

## Results
### EROS is a physiological regulator of gp91*phox*

EROS is essential for expression of the gp91*phox*-p22*phox* components of the phagocyte NADPH oxidase in primary human cells. We have previously shown this in patient-derived material but here also introduce an induced pluripotent stem cells (iPS)-derived macrophages rendered deficient in human EROS via CRISPR-Cas9 (*Figure 1—figure supplement 1A*). We examined the exact mechanism of action of EROS, asking how it fitted into the canonical model of gp91*phox*-p22*phox* biology (*Figure 1—figure supplement 1B*). One established method to investigate gp91*phox*-p22*phox* biosynthesis and stability is to transfect components of the complex into cells that do not normally express them (*Dinauer, 2019*). Such reductionist studies have demonstrated the ability of p22*phox* to stabilise the mature 91 kDa form of gp91*phox*. Co-expression of murine EROS and gp91*phox* resulted in markedly increased gp91*phox* expression relative to gp91*phox* transfection alone in NIH3T3 and COS-7 cells (*Figure 1A and B*), which are known not to express endogenous p22*phox* (*Yu et al., 1997*), as well as in HEK293T cells (*Figure 1C*), which express some endogenous p22*phox*. Thus, EROS can increase the expression of mouse gp91*phox* and can do so in the absence of p22*phox*.

This result was corroborated using human constructs of EROS, gp91*phox,* and p22*phox* (*Figure 1D–F*). In HEK293T cells, co-transfection of human EROS with gp91*phox* resulted in increased expression of the predominantly lower molecular weight (immature form) of gp91*phox* (*Figure 1D*). EROS's ability to enhance the expression of the 58 kDa form specifically was maintained when all three of gp91*phox*, p22*phox*, and EROS were transfected compared to gp91*phox* and p22*phox* only (*Figure 1E*). This effect was also readily observable in non-adherent HEK293-F cells and with different tags placed on EROS and gp91*phox* (*Figure 1—figure supplement 1C*). Similarly, in p22*phox*-deficient NIH3T3 cells, co-transfection of human EROS with gp91*phox* enhanced the abundance of the immature 58 kDa form (*Figure 1—figure supplement 1D*).

These data demonstrate the ability of EROS to increase immature 58 kDa gp91*phox* abundance in the absence of any p22*phox* in a manner that is, therefore, independent of p22*phox*'s impact on gp91*phox* maturation. In contrast to the results above, EROS co-expression with p22*phox* did not increase p22*phox* expression with either human (*Figure 1F*) or mouse (*Figure 1—figure supplement 1E*) constructs. Thus, the primary effect of EROS protein is on the abundance of gp91*phox*, not p22*phox*. Given that these two proteins are only stable as a heterodimer (*Segal, 1987*; *Dinauer et al., 1990*), EROS needs only be a critical regulator of one of them. Of note, EROS expression was normal in gp91*phox*-deficient cells (*Figure 1—figure supplement 1F*).

We next examined whether EROS affected stability of the 58 kDa form of gp91*phox* by using cycloheximide to block de novo protein synthesis. The 58 kDa protein was prominently stabilised by co-transfection with EROS with less degradation in its presence (*Figure 1G*, left panel). Hence,

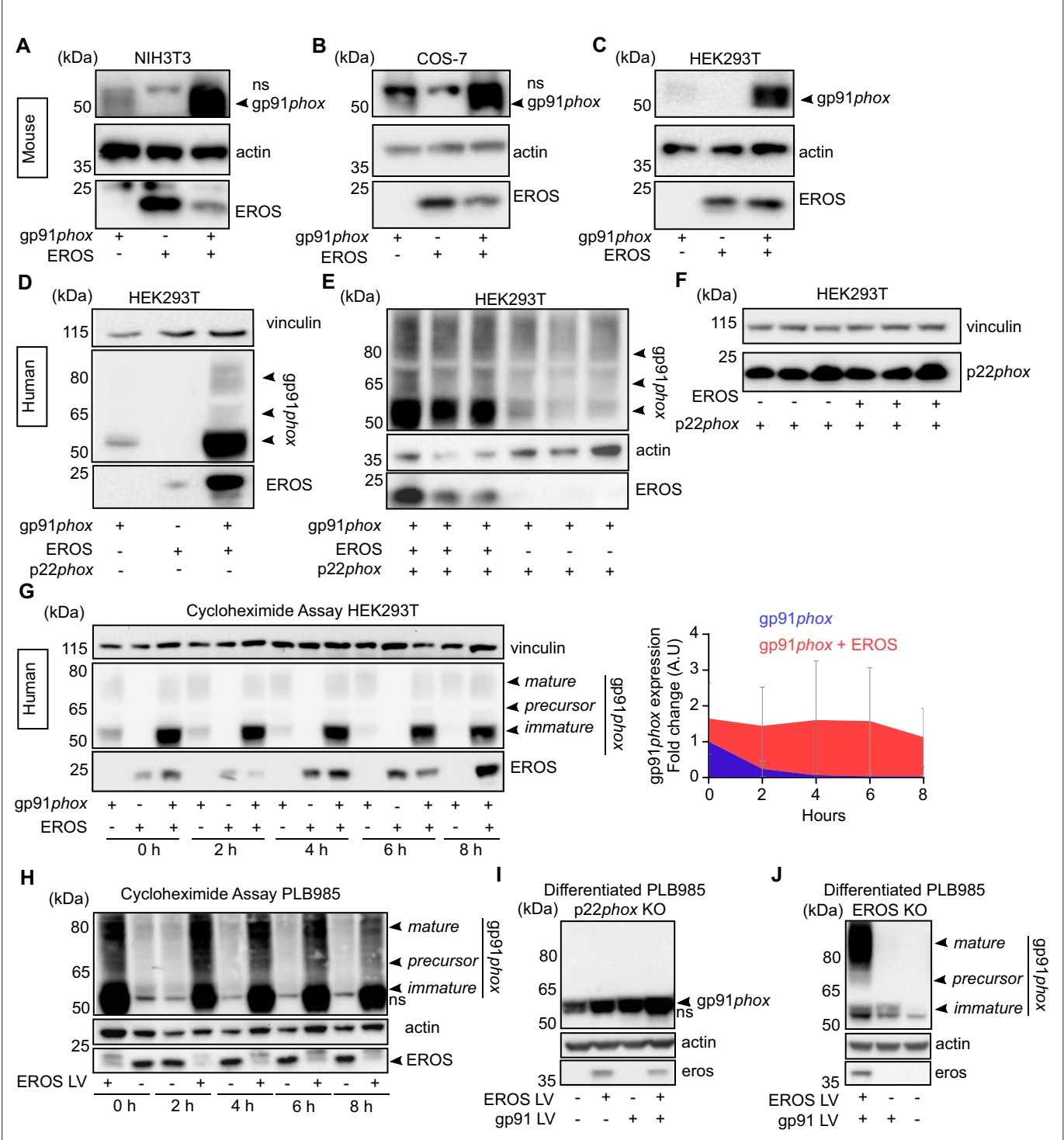

**Figure 1.** EROS stabilises the expression of gp91*phox* precursor. (**A–C**) Mouse constructs encoding EROS and gp91*phox* were co-transfected into NIH3T3 (**A**), COS-7 (**B**), and HEK293T (**C**) cell lines. gp91*phox* expression was analysed by immunoblotting; arrow indicates gp91*phox* band; ns: non-specific band. (**D–F**) gp91*phox* and p22*phox* expression in HEK293T cells following transfection with the indicated human constructs. (**G**) Left panel: analysis of the stability of the different forms of gp91*phox* (indicated by the arrows) following transfection in HEK293T cells in the presence or absence of EROS and treatment with 10 µg/mL cycloheximide. Right panel: quantitation of the cycloheximide assay (mean of four independent experiments; error bars indicate SD) represented as a fold change of gp91*phox* in cells expressing gp91*phox* and EROS vectors relative to gp91*phox* vector alone at 0 hr and normalised to actin expression. Actin and vinculin were used as loading control. (**H**) Stability of endogenous gp91*phox* in PLB985 neutrophil-like cells overexpressing lentivirus (LV) EROS-GFP vector (MW ≈ 41 kDa) and treated with 10 µg/mL cycloheximide. (**I–J**) gp91*phox* expression following

*Figure 1 continued on next page*

*Figure 1 continued*

lentiviral transduction of EROS-GFP, gp91*phox,* or both in differentiated PL985 knockout (KO) for p22*phox* (**I**) or EROS (**J**). Data are representative of three independent experiments. See also *Figure 1—figure supplement 1* and *Figure 1—source data 1–4*.

The online version of this article includes the following source data and figure supplement(s) for figure 1:

**Source data 1.** Raw unedited blots for *Figure 1A–C*.

**Source data 2.** Raw unedited blots for *Figure 1D–F*.

**Source data 3.** Raw unedited blots for *Figure 1G–J*.

**Source data 4.** Uncropped gels used for *Figure 1A–J*.

**Figure supplement 1.** EROS specifically regulates gp91*phox* not p22*phox* expression.

**Figure supplement 1—source data 1.** Raw unedited blots for *Figure 1—figure supplement 1A, C, and D*.

**Figure supplement 1—source data 2.** Raw unedited blots for *Figure 1—figure supplement 1E and F*.

**Figure supplement 1—source data 3.** Uncropped gels used for *Figure 1A, C–F*.

densitometry performed on the 58 kDa band (*Figure 1G*, right panel) resulted in a significantly greater area under curve in EROS and gp91*phox* co-transfection compared to gp91*phox* transfection alone (*Figure 1—figure supplement 1G*). The stability of the 65 kDa precursor form and the mature gp91*phox* was not affected by the presence of EROS (*Figure 1G* and data not shown).

We complemented this co-transfection work by examining PLB985 cells which express high levels of NADPH oxidase components, especially when they are differentiated into a mature neutrophil phenotype. Overexpression of EROS in PLB985 cells resulted in an increase of the abundance of endogenous gp91*phox*, particularly the 58 kDa form, which is barely detectable in control PLB985 cells (*Figure 1H*). Analysis of gp91*phox* expression in differentiated PLB985 cells deficient in p22*phox* showed minimal gp91*phox* expression but a small amount of the 58 kDa form could be observed. The abundance of this immature form can be increased by lentiviral overexpression of EROS or gp91*phox*, with highest abundance when both were transduced. Consistent with the absence of p22*phox*, the mature glycosylated form is not seen (*Figure 1I*). Differentiated PLB985 cells deficient in EROS also express minimal gp91*phox* at baseline. However, the small amount of gp91*phox* that can be detected following forced lentiviral overexpression is of both immature and mature forms because these cells are not genetically deficient in p22*phox* (*Figure 1J*).

These data emphasise the necessary role of EROS in stabilising the immature form of gp91*phox* in a human hematopoietic cell line, similar to that observed in our reductionist study.

## EROS can associate with the gp91*phox*-p22*phox* heterodimer through direct interaction with gp91*phox*

In our previous work, we showed that FLAG-tagged EROS could immunoprecipitate with gp91*phox*. Similarly, we found that endogenous gp91*phox* immunoprecipitated with endogenous EROS in PLB985 cell line (*Figure 2—figure supplement 1A*). We explored the details of the association between EROS, gp91*phox,* and p22*phox* using optimised co-immunoprecipitation conditions in HEK293-F cells expressing FLAG-tagged mouse EROS followed by size-exclusion chromatography (SEC) to further purify any complexes containing EROS. In cells transfected with EROS-FLAG, gp91*phox*-GFP, and p22*phox*, EROS co-immunoprecipitated with the partially glycosylated form of gp91*phox* (*Figure 2A*). SEC further showed that it could associate with the heme-bound form of gp91*phox* as evidenced by the co-elution with the heme absorbance peak (*Figure 2B*). Immunoblotting of SEC eluate detected p22*phox,* indicating that EROS forms a complex with both gp91*phox* and p22*phox* (*Figure 2C*). Notably, inhibiting heme synthesis using succinyl acetone blocked insertion of heme into gp91*phox* and prevented co-immunoprecipitation with p22*phox*, but did not prevent the association between EROS and gp91*phox* (*Figure 2D and E*).

These data suggested that while EROS increases abundance of and stabilises the immature form of gp91*phox*, it can remain associated with gp91*phox* as it binds heme and then p22*phox*. Thus, we asked whether EROS interacts directly with gp91*phox*. To address this, we utilised the Nanoluc Binary Technology (NanoBiT) complementation reporter system (*Dixon et al., 2016*). We fused EROS and gp91*phox* to large BiT (LgBiT) and small BiT (SmBiT) subunits of the luciferase at either the N- or C-terminus. Constructs were transfected into HEK293T to test various possible combinations

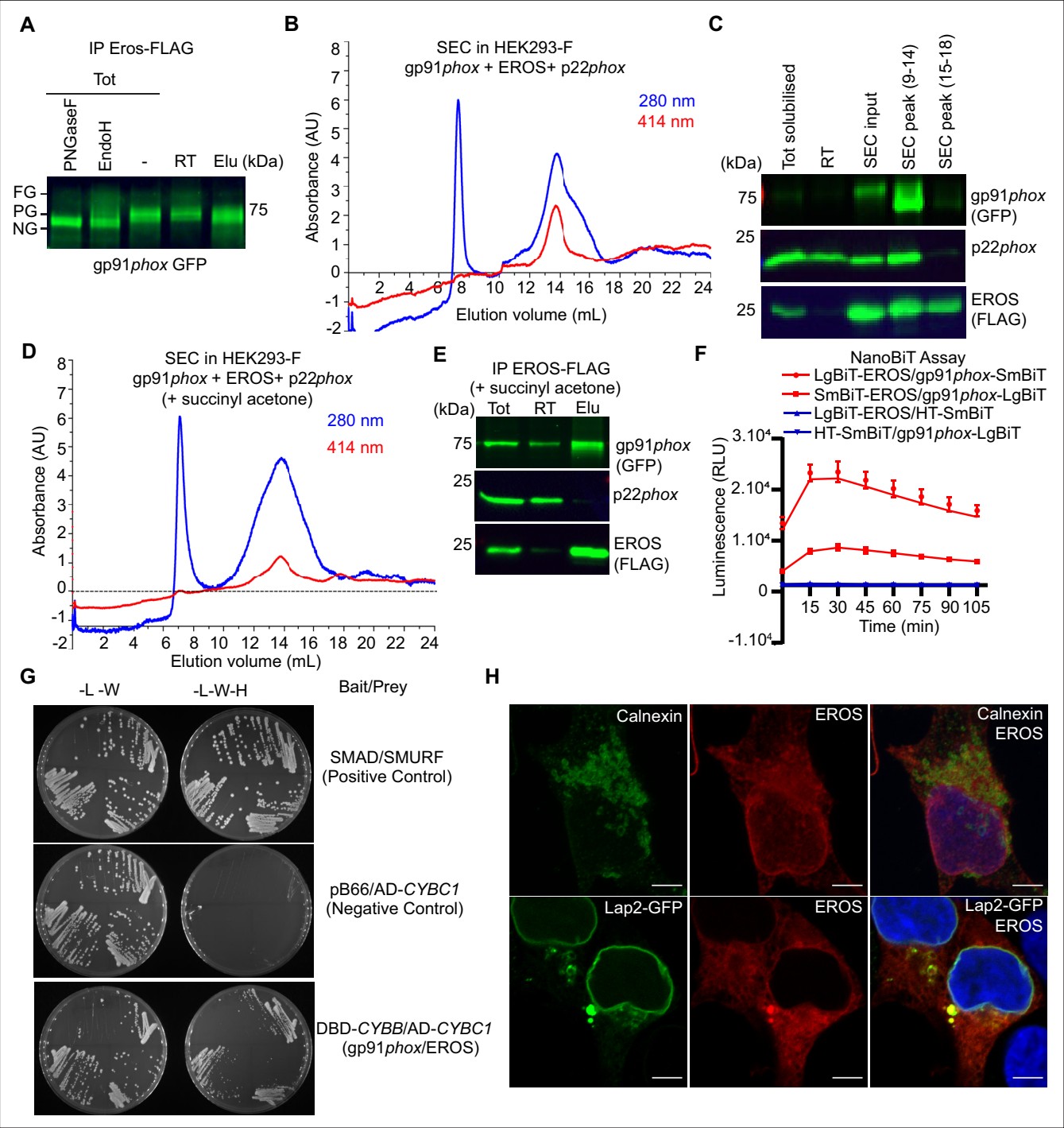

**Figure 2.** EROS regulates flavocytochrome b formation via direct binding to gp91*phox*. (**A–D**) Immunoprecipitation (IP) and size-exclusion chromatography (SEC) analysis of protein complexes associated with EROS. (**A**) IP of EROS in HEK293-F cells expressing StrepII-FLAG-tagged EROS, gp91*phox*-GFP, and p22*phox* with Western blot for gp91*phox*. Lysates treated with peptide N-glycosidase F (PNGaseF) or endoglycosidase H (EndoH) served as reference; FG: fully glycosylated; PG: partially glycosylated; NG: non-glycosylated; Tot: total lysate; RT: run through; Elu: eluate. (**B**) SEC profile of EROS-IP eluate indicating protein (280 nm) and heme (414 nm) content. (**C**) Immunoblot analysis of gp91*phox*-GFP, EROS-FLAG, and endogenous p22*phox* in SEC fractions 9–14 and 15–18. (**D**) SEC profile of EROS eluate from HEK293-F cells expressing EROS-FLAG, gp91*phox*, and p22*phox* constructs and treated with heme biosynthesis inhibitor succinyl acetone (10 µg/ml). (**E**) IP of StrepII-FLAG-tagged EROS in HEK293-F treated with succinyl acetone. (**F**) Interaction between gp91*phox* and EROS assessed through luminescence production in live HEK293T cells expressing the indicated plasmids fused with the large (LgBIT) or small (SmBIT) fragment of the NanoLuc luciferase (see 'Methods'). Halo Tag (HT)-SmBIT is the negative control; RLU: relative luminescence unit. (**G**) Yeast growth phenotypes obtained with the specified selective media using gp91*phox* bait

*Figure 2 continued on next page*

*Figure 2 continued*

plasmid and EROS prey plasmid. L: leucine; W: tryptophan; H: histidine; DBD: DNA binding domain of Gal4; AD: activation domain of Gal4 (see 'Methods'). (**H**) EROS localisation in HEK293 cells transfected with EROS construct (top panel; 3D stack) or EROS and Lap2-GFP constructs (bottom panel; single plane), fixed, permeabilised, and labelled with anti-EROS and anti-calnexin antibodies. Scale bar = 5 µm. Data are representative of at least three independent experiments; error bars indicate SEM of triplicates. See also *Figure 2—figure supplement 1* and *Figure 2—source data 1–2*.

The online version of this article includes the following source data and figure supplement(s) for figure 2:

**Source data 1.** Raw unedited blots for *Figure 2A, C and E*.

**Source data 2.** Uncropped gels used for *Figure 2*.

**Figure supplement 1.** EROS acts at the early stage of gp91*phox* biosynthesis.

**Figure supplement 1—source data 1.** Raw unedited blots for *Figure 2—figure supplement 1A and D*.

**Figure supplement 1—source data 2.** Raw unedited blots for *Figure 2—figure supplement 1E–G*.

**Figure supplement 1—source data 3.** Raw unedited blots for *Figure 2—figure supplement 1I*.

**Figure supplement 1—source data 4.** Uncropped gels used for *Figure 2—figure supplement 1A,D–F*.

**Figure supplement 1—source data 5.** Uncropped gels used for *Figure 2—figure supplement 1G–I*.

**Figure supplement 2.** Diagram depicting the role of EROS in gp91*phox* biosynthesis and formation of the heterodimer with p22*phox*.

(*Figure 2—figure supplement 1B*). The LgBiT-EROS and gp91*phox*-LgBiT constructs paired with the negative control HaloTag-SmBiT (HT-SmBiT) gave signal within the background. Two separate construct pairs, LgBiT-EROS with gp91*phox*-SmBiT and SmBiT-EROS with gp91*phox*-LgBiT, generated high-intensity luminescence within the first 30 min of monitoring live HEK293T cells (*Figure 2F*). The combination LgBiT-EROS and gp91*phox*-SmBiT encoded in a single vector (BiBiT vector, see 'Methods') where expression of both gp91*phox* and EROS is driven by the same promotor also generated high-intensity luminescence (*Figure 2—figure supplement 1C*), thereby demonstrating that gp91*phox* and EROS interact directly. This construct, where EROS and gp91*phox* are encoded by a single vector, provides a convenient platform to interrogate questions such as the effect of missense mutations in either protein on their binding or whether certain small molecules can disrupt the interaction.

**Table 1.** Yeast 2 Hybrid interaction matrix.

| Interaction matrix | | | Selection medium | | |
|---|---|---|---|---|---|
| **Type** | **Bait** | **Prey** | **DO-2** | **DO-3** | **DO-3 + 0.5 mM 3-AT** |
| Positive control | SMAD | SMURF | + | + | / |
| Negative control | pB66Ø | AD-*CYBC1* | + | - | - |
| Negative control | pB66Ø | AD-*CYBB* | + | - | / |
| Negative control | DBD-*CYBC1* | pP7Ø | + | - | / |
| Negative control | DBD-*CYBB* | pP7Ø | + | - | - |
| Interaction | DBD-*CYBC1* | AD-*CYBB* | + | - | / |
| Interaction | DBD-*CYBB* | AD-*CYBC1* | + | + | +/- |
| Interaction | DBD-*CYBC1* | AD-*CYBC1* | + | - | / |
| Interaction | DBD-*CYBB* | AD-*CYBB* | + | - | / |

Table resuming the different conditions tested during probing of interaction between EROS (*CYBC1*) and gp91*phox* (*CYBB*). pB66: Gal4 DNA-Binding Domain (DBD) vector, i.e. bait vector (DBD-bait); pB66ø: empty pB66 vector; pP7: Gal4 Activation Domain (AD) vector, i.e. prey vector (AD-prey). The same AD protein is expressed from both plasmids; pP7ø: empty pP7 vector; DBD-*CYBC1*: aa 1–187 of EROS cloned into pB66. Hybrigenics' reference for this construct is hgx4414v2_pB66; DBD-*CYBB*: aa 1–570 of gp91*phox* cloned into pB66. Hybrigenics' reference for this construct is hgx5346v1_pB66; AD-*CYBC1*: aa 1–187 of EROS cloned into pP7. Hybrigenics' reference for this construct is hgx4414v2_pP7; AD-*CYBB*: aa 1–570 of gp91*phox* cloned into pP7. Hybrigenics' reference for this construct is hgx5346v1_pP7; DO-2: selective media without tryptophan and leucine. DO-3: selective media without tryptophan, leucine and histidine. 3-AT: 3-aminotriazole (see 'Methods').

This is further supported by Yeast 2 Hybrid experiments. Using gp91*phox* bait plasmid and EROS prey plasmid (*Table 1*), we specifically observed colonies under the selective media without leucine, tryptophan, and histidine (see 'Methods'), which confirmed the direct interaction between EROS protein and gp91*phox* (*Figure 2G*, *Figure 2—figure supplement 2H*). Confocal microscopy analysis showed that EROS is found in the ER and perinuclear compartment (*Figure 2H*), colocalising with calnexin and the lamina-associated polypeptide-2 (Lap2; a nuclear membrane marker). This result is consistent with previous literature reporting a nuclear membrane localisation of the uncharacterised protein C17ORF62, which corresponds to human EROS (*Korfali et al., 2010*). Given the continuity between the nuclear membrane and the ER, these data suggest that EROS acts very early in gp91*phox* biosynthesis.

These findings support a model where EROS directly binds and stabilises the 58 kDa gp91*phox* protein, remaining associated with it through heme incorporation and p22*phox* binding. The role of EROS is thus distinct and temporally separated from that of p22*phox*. In the ER, p22*phox* interacts with gp91*phox* after heme has been incorporated and stabilises the partially glycosylated form (*Figure 2—figure supplement 2*). Notably, in the HEK293 co-transfection system, we found that human EROS increased expression of NOX1 and NOX4, two close homologues of gp91*phox* (NOX2). For NOX1, NOX2, and NOX4, this process was not impaired by succinyl acetone treatment (*Figure 2—figure supplement 1D–F*). EROS was not able to increase NOX5 expression, which is structurally different from NOX1, 2, 4 (*Figure 2—figure supplement 1G*). A further Yeast 2 Hybrid experiment, using a separate method, as described in *Luck et al., 2020* demonstrated direct binding of EROS to NOX2-3-4 but not NOX5 or p22*phox* (*Figure 2—figure supplement 2H*). However, Western blot analysis of whole kidney and heart lysate from EROS knockout mice showed no difference in nox4 expression in EROS knockout relative to control (*Figure 2—figure supplement 1I*). Thus, EROS does not regulate nox4 physiologically in the kidney and heart. We also assayed colon from control and EROS knockout mice for nox1 expression using the highly specific antibody described in *Diebold et al., 2019*. We did not see a difference in expression between control and EROS knockout mice (*Figure 2—figure supplement 1I*). Non-specific bands were seen with EROS antibody (which is common in mouse lysates) but we could not detect EROS band which is consistent with its low expression level in these tissues.

Nevertheless, our co-transfection and Yeast 2 Hybrid data may provide important clues to the motifs that EROS recognises in directly binding NOX proteins.

## EROS-mediated regulation of gp91*phox* and the OST complex

The ability of EROS to act at the earliest stage of gp91*phox* biosynthesis prompted us to examine its biology in greater detail. To identify EROS-associated proteins, we performed four biological replicate FLAG affinity purifications from both RAW 264.7 macrophages overexpressing FLAG-tagged EROS and untagged control cells (*Thomas et al., 2017b*). We used the widely adopted statistical algorithm Significance Analysis of INTeractome (SAINTexpress; *Teo et al., 2014*) to discriminate specific EROS interactors from non-specific or background binders. SAINTexpress uses quantitative information from the mass spectrometry data, more specifically the abundance of each identified protein in the bait experiments compared to the control purifications, to score the probability of a true interaction. The SAINT probability (SP) score represents the confidence level of potential protein interactions, with 1 being the highest possible value. We applied a stringent threshold of SP > 0.9 (false discovery rate [FDR] < 1%) to derive a list of 59 high-confidence EROS interactors (*Figure 3A*). Network analysis using STRING interaction database showed that the majority of these proteins were connected between them through physical interactions (*Figure 3B*). Gene Ontology (GO) enrichment analysis revealed 'N-linked glycosylation' as one of the most enriched terms in our EROS-interacting protein set, and 'positive regulation of glycoprotein biosynthetic process' was also enriched. High-confidence EROS interactors OST 48 kDa subunit (DDOST/OST48), ribophorin-1 (RPN1), and ribophorin 2 (RPN2) form the non-catalytic subunit of the OST complex (*Pfeffer et al., 2014*). Staurosporine and temperature-sensitive 3A (STT3A), which forms the catalytic subunit of the OST complex (*Ramírez et al., 2019*), was specifically identified in three out of four EROS pull-down experiments and not in controls, although it did not make the strict SP score cut-off, suggesting that it might also be an EROS interactor. These data point to EROS being bound to gp91*phox* while N-glycosylation takes place.

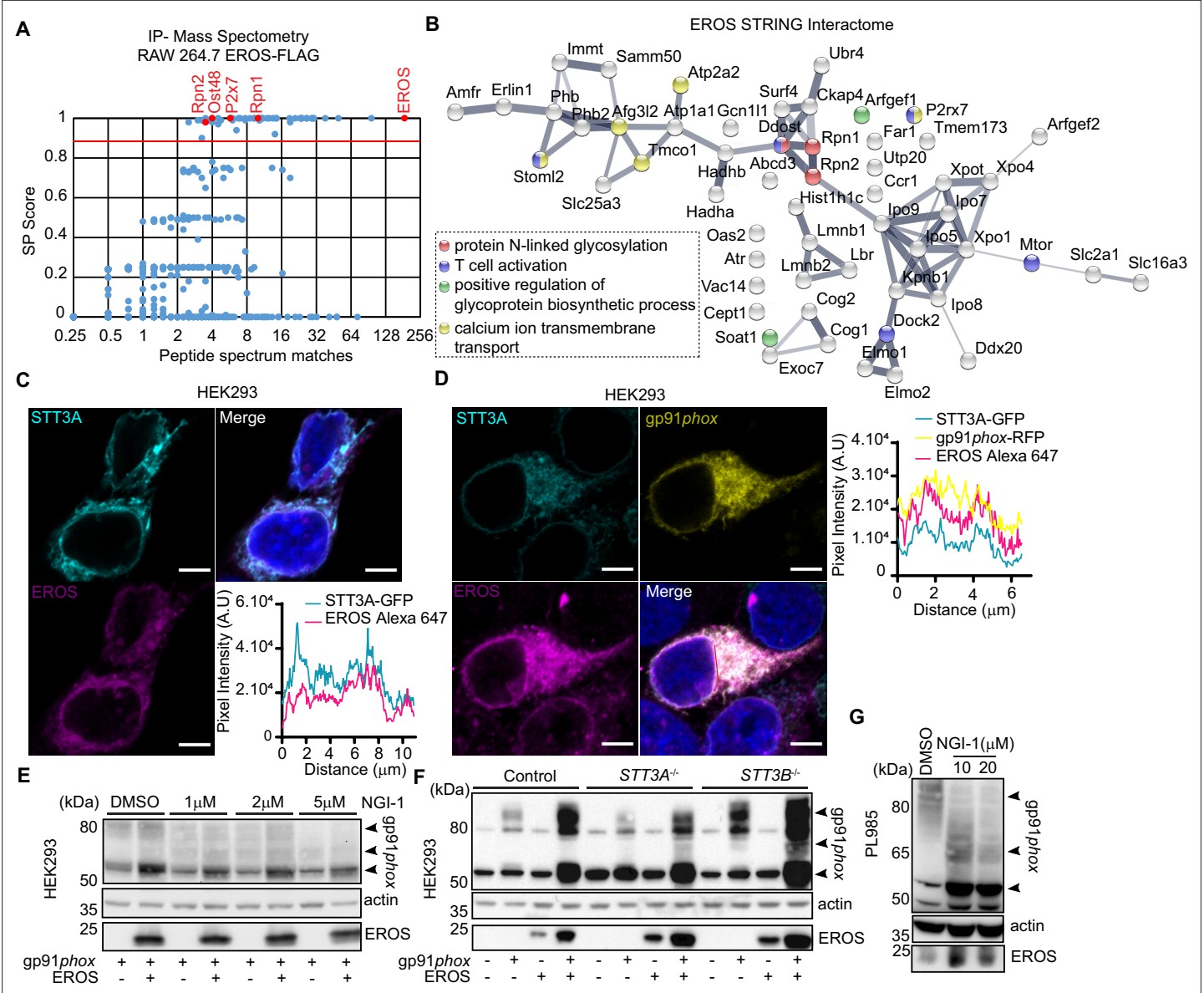

**Figure 3.** EROS interacts and colocalises with the oligosaccharyltransferase (OST) complex. (**A**) EROS-FLAG affinity purification-mass spectrometry (AP-MS). Graph showing abundance (average number of peptide spectrum matches across four biological replicates) of all proteins identified in the FLAG AP-MS experiments (blue and red dots) versus their interactor specificity (SAINT probability score: SP). The red line marks the SP score cut-off (0.9) for high-confidence interacting proteins. Proteins (dots) above this cut-off (59) are deemed high-confidence interactors. The bait (EROS) and interacting proteins relevant to this study are shown in red. (**B**) Protein interaction network of the 59 high-confidence EROS-interacting proteins (SP >0.9). The protein interactions were derived from STRING. Coloured nodes represent proteins annotated with enriched Gene Ontology (GO) terms relevant to this study. (**C, D**) EROS and gp91*phox* localization, following fixation and labelling with anti-EROS antibody, in HEK293 cells expressing STT3A-GFP and EROS constructs (**C**) or STT3A-GFP, gp91*phox*-mRFP and EROS untagged constructs (**D**); scale bars = 5 µm. Graphs represent the intensity profile of STT3A-GFP and EROS signal or STT3A-GFP, gp91*phox*-mRFP, and EROS signal measured across the nuclear membrane (indicated in red line). (**E**) Expression of gp91*phox* in HEK293 cells transfected with the indicated constructs and treated with OST inhibitor (NGI-1) at the indicated concentration. (**F**) Expression of gp91*phox* in control and *STT3A*[-/-] or *STT3B*[-/-] HEK293 cells transfected with the indicated vectors. (**G**) Expression of gp91*phox* in PLB985 cell line treated with NGI-1 at the indicated concentration. Data are representative of three independent experiments. See also ***Figure 3— figure supplement 1*** and ***Figure 3—source data 1–3***.

The online version of this article includes the following source data and figure supplement(s) for figure 3:

**Source data 1.** Table of the 59 proteins identified in EROS IP-MS interactome.

**Source data 2.** Raw unedited blots for ***Figure 3E–G***.

**Source data 3.** Uncropped gels used for ***Figure 3E–G***.

*Figure 3 continued on next page*

*Figure 3 continued*

**Figure supplement 1.** Analysis of gp91*phox* expression and ER-stress marker in different cell lines.

**Figure supplement 1—source data 1.** Raw unedited blots for *Figure 3—figure supplement 1A–D*.

**Figure supplement 1—source data 2.** Uncropped blots for *Figure 3—figure supplement 1A–D*.

Translation, synthesis, and N-glycosylation of proteins transferred into the ER involve the ER translocon-machinery comprising the SEC61 channel, the translocon-associated protein (TRAP) complex, and the OST complex responsible for the addition of N-linked oligosaccharides to nascent protein (*Cherepanova et al., 2016*). We hypothesised that EROS may cooperate with the OST complex to facilitate the stabilisation and maturation of gp91*phox*. HEK293 cells expressing STT3A-GFP and EROS-mRFP constructs exhibited an ER localisation of the two proteins with a specific rim around the nucleus (*Figure 3C*, right panel). Quantitation of the pixel intensity of STT3A-GFP and EROS-mRFP across the nuclear membrane gave an identical profile (*Figure 3C*, left panel). Similarly, co-expression of STT3A-GFP, gp91*phox*-mRFP, and EROS untagged constructs lead to an ER and perinuclear localisation of the three proteins (*Figure 3D*, right panel) with an identical pixel intensity profile upon quantitation (*Figure 3D*, left panel). These data demonstrate that a proportion of gp91*phox* can be found at the site of STT3A and EROS localisation. Inhibition of the OST complex with the selective compound NGI-1 (*Puschnik et al., 2017*; *Rinis et al., 2018*) impaired the ability of EROS to increase the immature 58 kDa form in a dose-dependent manner (*Figure 3E*) in HEK293 cells.

To complement these findings, we expressed the gp91*phox* construct alone or in combination with the EROS construct in HEK293 deficient in *STT3A* or *STT3B* (*Cherepanova and Gilmore, 2016*). STT3A interacts directly with SEC61 and mediates co-translational N-glycosylation. STT3B is not associated with SEC61, and glycosylates sites skipped by STT3A. In the absence of STT3A, we observed a shift in the glycosylation pattern of gp91*phox* compared to control conditions, resulting in an increase of the immature 58 kDa form (*Figure 3F*). Co-expression of gp91*phox* with EROS in *STT3A*$^{-/-}$ cells resulted in the same effect with the appearance of further bands around 65 kDa, probably glycosylation intermediates. *STT3B*$^{-/-}$ cells, instead, exhibited an overall increase of gp91*phox* abundance compared to control cells both in the absence or presence of EROS (*Figure 3F*). These glycosylation intermediates were suppressed by treatment of *STT3A*$^{-/-}$ cells expressing gp91*phox* and EROS with the inhibitor NGI-1, leaving only the 58 kDa band (*Figure 3—figure supplement 1A*). The 58 kDa band was not detectable in cells treated with the compound tunicamycin, a widely used N-glycosylation inhibitor. The overall increase of gp91*phox* abundance in *STT3B*$^{-/-}$ cells in the presence of EROS was also impaired by treatment with NGI-1 (*Figure 3—figure supplement 1A*). Interestingly, NGI-1 treatment of PLB985 cells also shifted the glycosylation pattern of gp91*phox*, selectively enhancing bands at 65 kDa and 58 kDa form compared to DMSO treatment where we observed a predominantly mature 91 kDa protein (*Figure 3G*). This suggests that blockade of the OST machinery in the ER prevents subsequent proper glycosylation in the Golgi apparatus. Lack of STT3A as well as treatment with NGI-1 led to an increase in the expression of the ER stress marker BiP in HEK293 and in PLB985 cells (*Figure 3—figure supplement 1B–D*). This finding is recapitulated by treatment of PLB985 cells with tunicamycin (*Figure 3—figure supplement 1D*).

Overall, these findings are consistent with EROS being bound to gp91*phox* at the early stages of its biosynthesis while N-glycosylation takes place in ER. They also reveal a previously unappreciated involvement of the OST complex in gp91*phox* maturation process stressing a requirement of the catalytic subunit STT3A in the co-translational glycosylation of gp91*phox*.

## EROS targets a specific group of proteins

Having defined the role of EROS in gp91*phox*-p22*phox* heterodimer maturation, we leveraged new artificial intelligence tools to gain insights into its structure. The structure of the full-length EROS protein predicted by AlphaFold 2.0 (*Jumper et al., 2021*) is available through the AlphaFold Protein Structure Database (*Tunyasuvunakool et al., 2021*). The predicted structure shows a tripartite organisation of (i) a PH domain, (ii) an integral membrane segment, and (iii) a flexible C-terminal tail (*Figure 4A*). The EROS protein contains a Pleckstrin Homology (PH) domain composed of residues 1–20 (containing beta strands 1 and 2) and residues 62–166 (containing beta strands 3–8 and alpha helices H3 and H4). Inserted between beta strands 2 and 3 of the PH domain are two integral membrane alpha

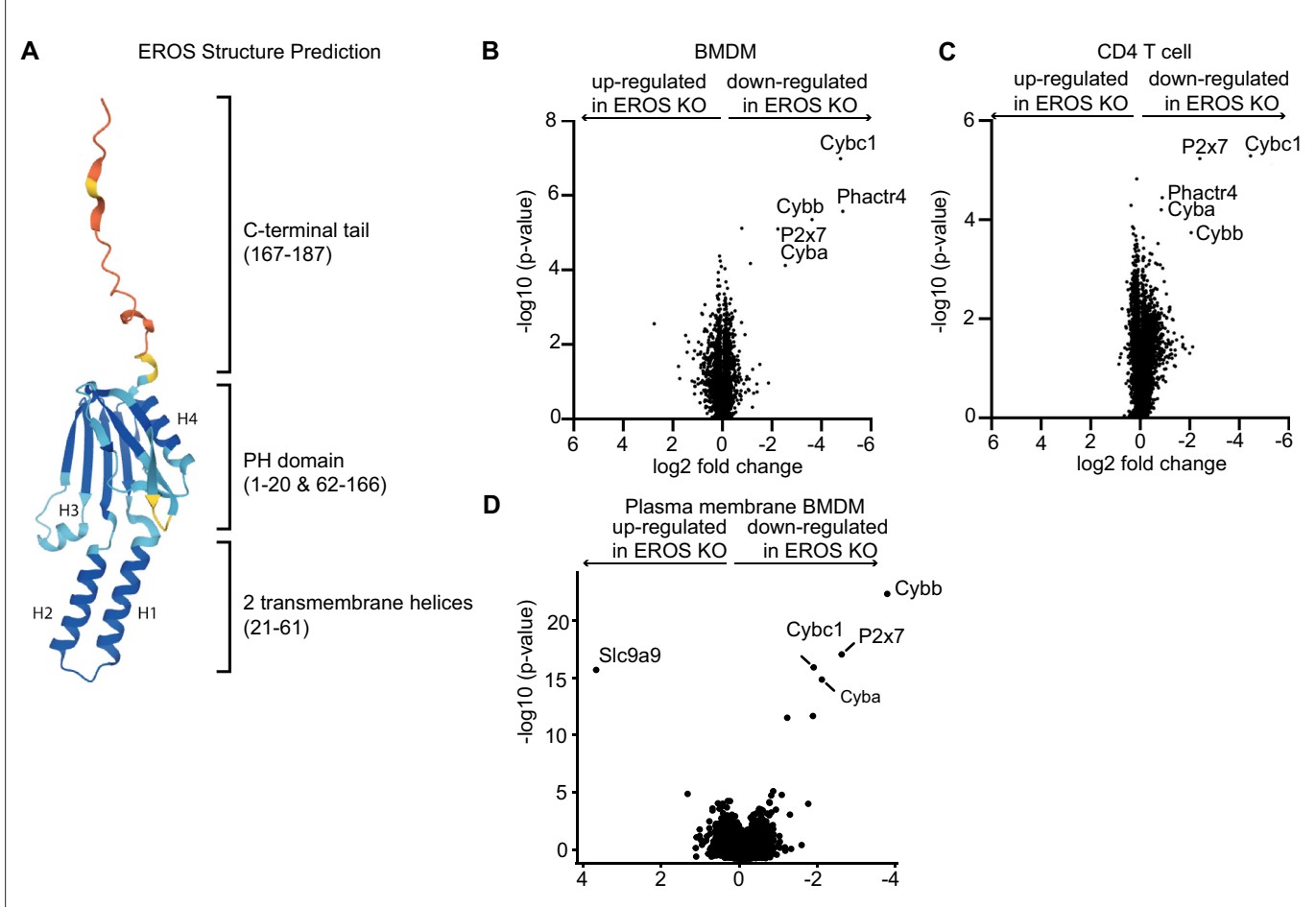

**Figure 4.** Mass spectrometry analysis of proteins modulated by EROS in immune cells identified the P2X7 purinergic receptor. (**A**) Cartoon representation of the AlphaFold structure prediction of the top ranked model. The structure is coloured according to the model quality with dark blue representing residues with a predicted local Distance Difference Test (plDDT) score > 90, light blue plDDT > 70, yellow plDDT > 50, and red plDDT ≤ 50. (**B–D**) Volcano plot of proteins detected by Tandem Mass Tagging proteomics analysis of bone marrow-derived macrophages: BMDM (**B**) and CD4⁺ T lymphocytes (**C**) isolated from control and EROS knockout (KO) mice. (**D**) Volcano plot of proteins recovered by Plasma Membrane Profiling of macrophages isolated from control and EROS KO mice. The volcano plots display statistical significance (-log10 p-value) versus the log2 fold change from five biological replicates.

helices (helices H1 and H2) that are packed in an antiparallel orientation. The C-terminal tail of EROS composed of residues 167–187 has a structure with very low confidence <50 plDDT (predicted local Distance Difference Test) which is usually indicative of disordered regions of proteins. Multiple independent models built with AlphaFold (data not shown) show a wide variety of conformations for this tail confirming its lack of regular structure. If confirmed experimentally, the insertion of transmembrane helices within a PH domain appears to be a hitherto unobserved feature within known structures. A similar PH domain organisation is predicted to be found in the structure of the Phosphatidylinositol Glycan Anchor Biosynthesis class H (PIGH) protein (UniProtKB-Q5M9N4). This ER-associated transmembrane protein is part of the glycosylphosphatidylinositol-N-acetylglucosaminyltransferase (GPI-GnT) complex that catalyses the transfer of N-acetylglucosamine from UDP-N-acetylglucosamine to phosphatidylinositol and participates in the first step of GPI biosynthesis. GPI is an anchor for many membrane proteins, including key immune proteins such as the receptors CD16 and CD14 (*Wegner et al., 2021*), as well as the complement regulatory proteins CD55 and CD59 (*Tremblay-Laganière et al., 2021*). Therefore, we asked whether EROS might regulate the abundance of other proteins in different immune cell types.

We performed Tandem Mass Tagging proteomics analysis on bone marrow-derived macrophages (BMDM) and CD4 T cells from control and EROS knockout mice. This approach allowed the

identification of around 8000 proteins, a greater number than our previously reported label-free experiment which was limited to 2000 proteins (*Figure 4B and C*). We found that EROS deficiency has major effects on the expression of a small number of proteins in macrophages, and this effect is conserved in CD4 T cells. These proteins were gp91*phox* (*Cybb*), p22*phox* (*Cyba*), and P2x7, the latter also identified as a high-confidence EROS-interacting protein (see *Figure 3A*). Additionally, we observed a lower expression of the tumour suppressor protein Phactr4 (*Solimini et al., 2013*) in EROS-deficient cells. Phactr4 plays a key role in actin cytoskeleton remodelling by regulating the β1 integrin–FAK–ROCK-cofilin pathway (*Huet et al., 2013*; *Sun and Fässler, 2012*). The immunological role of Phactr4 is currently unknown.

Given that EROS seemed to affect proteins that localised to the plasma membrane, we performed plasma membrane profiling analysis to determine whether evaluating this compartment alone might reveal more proteins regulated by EROS. This analysis demonstrated very similar results. gp91*phox*, p22*phox*, and P2x7 were the most downregulated proteins in EROS-deficient cells compared to control cells. The sodium/proton exchanger Slc9a9 was significantly up-regulated (*Figure 4D*).

Thus, the biological effects of EROS deficiency are conserved across different immune cells and targets a very selective group of transmembrane proteins.

## EROS regulates the abundance of the P2X7 ion channel in mouse and human cells

The purinergic receptor P2X7 was consistently downregulated in all our mass spectrometry data (*Thomas et al., 2017b*, *Figure 4*) of EROS-deficient cells and was identified as a confident EROS interactor (*Figure 3A*). Proteins including Elmo1, Dock2, and mTOR (*Figure 3B*) are also confident EROS interactors, but their expression remained unchanged in EROS-deficient cells (*Figure 4B and C*). Thus, P2X7 seems the only confident EROS interactor (*Figure 3A and B*) whose expression is regulated by EROS similarly to gp91*phox*. Therefore, we focused on validating EROS's role in controlling P2X7 abundance. We verified this by Western blot (*Figure 5A*) and flow cytometry of both BMDM (*Figure 5—figure supplement 1A*) and freshly isolated peritoneal macrophages (*Figure 5—figure supplement 1B*) in independent cohorts of mice. The lack of P2X7 in EROS-deficient cells was not a consequence of lack of expression of the NADPH oxidase as P2X7 protein was expressed normally in gp91*phox*-deficient cells (*Figure 5B*). Consistent with our mass spectrometry data (*Figure 4B*), P2X7 expression was extremely low on CD4[+] T cells from EROS knockout mice, measured by Western blot and flow cytometry (*Figure 5—figure supplement 1C and D*). Control splenocytes had high levels of P2X7 expression in NKT cells andγδT cells with much reduced expression in EROS-deficient cells (*Figure 5—figure supplement 1E and F*). EROS deficiency also led to reduced P2X7 expression in B cells, NK cells, and CD8[+] T cells (*Figure 5—figure supplement 1G–I*).

P2X7 was very lowly expressed in human iPS-derived macrophages (*Figure 5C*) and PLB-985 cells (*Figure 5D*) that carried a CRISPR-mediated deletion in EROS. RAW264.7 macrophages express P2X7 endogenously and lentiviral overexpression of EROS increased P2X7 expression relative to that observed with a control vector that expressed only GFP (*Figure 5E*). This effect was reproduced in HEK293 cells. Co-transfection of EROS and P2X7 resulted in much higher expression of P2X7 than when P2X7 was transfected alone (*Figure 5F*, *Figure 5—figure supplement 1J*). Therefore, not only does EROS deficiency led to a lack of P2X7 expression, but EROS upregulation increased P2X7 abundance. We hypothesised that the regulation of P2X7 abundance would have a similar mechanism to that for the gp91*phox*-p22*phox* heterodimer. Accordingly, P2X7 co-immunoprecipitated with EROS from RAW264.7 macrophages that expressed an N-terminal FLAG-tagged EROS (*Figure 5G*). Furthermore, NanoBiT analysis using a LgBiT-EROS and P2X7-SmBiT pair constructs in HEK293 resulted in luminescence production, showing that the interaction between P2X7 and EROS was direct (*Figure 5H*).

Macrophages express other P2X family members including P2X4 and P2X1 (*Sim et al., 2007*). Western blot analysis of BMDM also demonstrated a lower expression of P2X1 in EROS-deficient cells (*Figure 5I*). As with P2X7, co-transfection of both P2X1 and EROS caused greater expression of P2X1 compared to when P2X1 was transfected alone (*Figure 5J*). Thus, EROS also regulates P2X1 protein abundance. This was not, however, the case for P2X4 (*Figure 5—figure supplement 1K*) which is mainly found in endolysosomal compartment in macrophages (*Boumechache et al., 2009*; *Robinson and Murrell-Lagnado, 2013*). These data emphasise the role of EROS in selectively controlling P2X7 expression through a direct recruitment analogous to that observed for gp91*phox*.

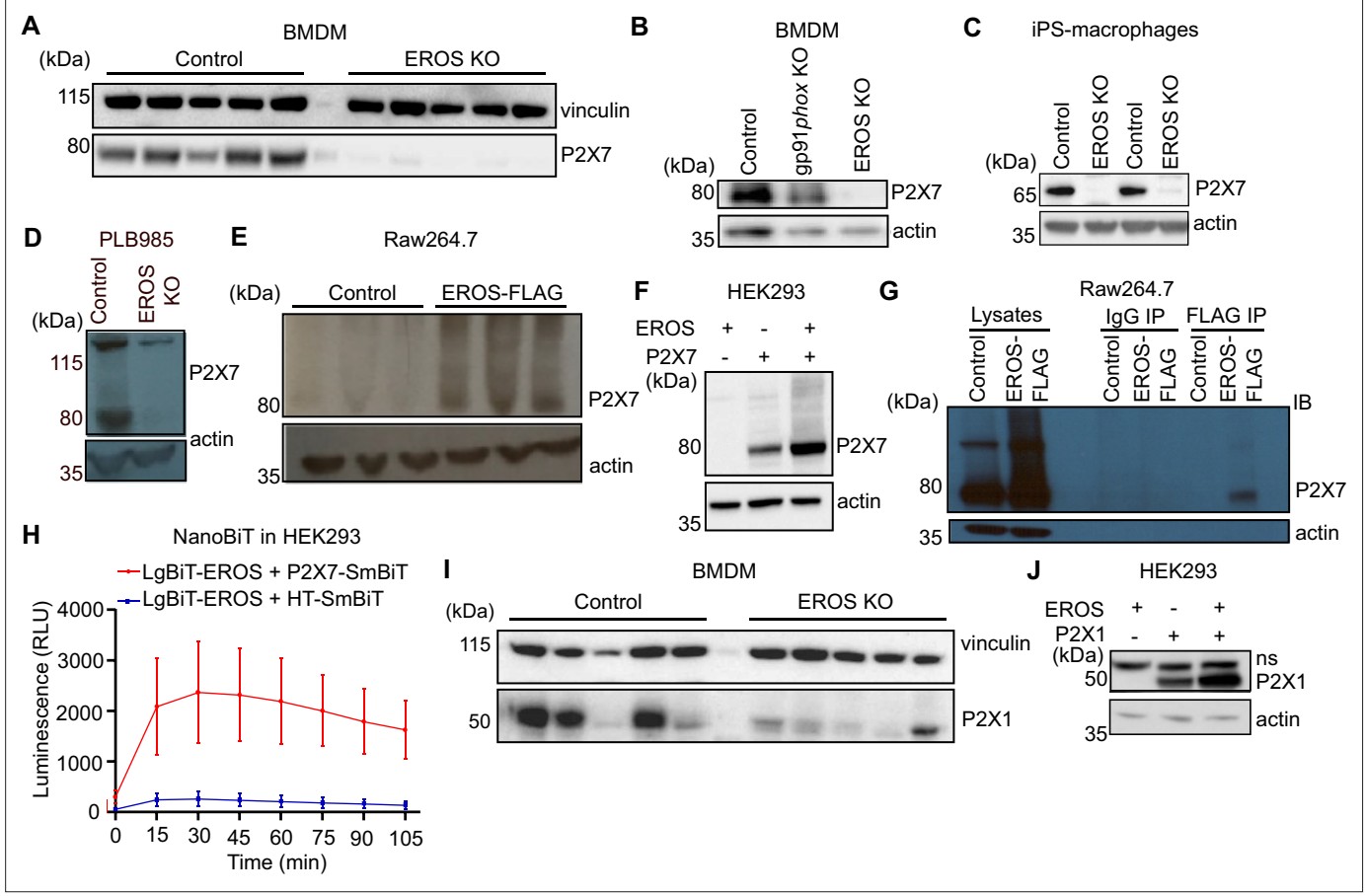

**Figure 5.** EROS regulates P2X7 protein abundance by direct interaction and independently of the nicotinamide adenine dinucleotide phosphate (NADPH) oxidase. (**A–D**) P2X7 expression analysed by Western blotting of macrophages isolated from control, EROS knockout (KO) (**A**) and gp91*phox* KO mice (**B**), induced pluripotent stem cells (iPS)-derived macrophages control or EROS-deficient (**C**) and of control PLB985 cells and an EROS-deficient clone (**D**). (**E, F**) P2X7 expression in RAW264.7 cells overexpressing a FLAG-tagged EROS vector (**E**) and in HEK293 cells transiently expressing the specified constructs (**F**). (**G, H**) Interaction between EROS and P2X7 probed by immunoprecipitation (IP) of EROS from RAW264.7 EROS-FLAG macrophages followed by immunoblot (IB) for P2X7 (**G**) and by Nanoluc Binary Technology (NanoBIT) assay in live HEK293 cells expressing the LgBIT-fused EROS vector with a SmBIT-fused P2X7 vector (**H**). (**I**) P2X1 expression in macrophages isolated from EROS KO mice compared to control. n = 5 biological replicates. (**J**) P2X1 abundance upon co-transfection with EROS construct in HEK293 cells. Data are representative of hree independent experiments; error bars indicate SEM of triplicates. See also *Figure 5—figure supplement 1* and *Figure 5—source data 1–4*.

The online version of this article includes the following source data and figure supplement(s) for figure 5:

**Source data 1.** Raw unedited blots for *Figure 5A–D*.

**Source data 2.** Raw unedited blots for *Figure 5E–G*.

**Source data 3.** Raw unedited blots for *Figure 5I and J*.

**Source data 4.** Uncropped gels used for *Figure 5A–G, I, J*.

**Figure supplement 1.** P2X7 expression is lower in numerous cell subsets from EROS knockout (KO) mice.

**Figure supplement 1—source data 1.** Raw unedited blots for *Figure 5—figure supplement 1C and K*.

**Figure supplement 1—source data 2.** Uncropped gels used for *Figure 5—figure supplement 1C and K*.

## Loss of P2X7 impairs inflammasome activation and shedding of surface ligand

P2X7 is a metabotropic ion channel and ligation by adenosine-5'-triphosphate (ATP) is associated with an inward flux of calcium and sodium an outward flux of potassium ions (*Bartlett et al., 2014*). We measured calcium influx following treatment of control and EROS-deficient BMDM by 2′(3′)-O-(4-Benzoylbenzoyl)ATP (bzATP), which has higher affinity than ATP for the P2X7 receptor. bzATP-driven calcium flux was highly attenuated in EROS-deficient macrophages compared to their

control counterpart (*Figure 6A*). Ionomycin-driven responses were equivalent between the two strains, showing that there was no inherent problem with calcium mobilisation in EROS deficiency (*Figure 6A*). Ligation of P2X7 by ATP is a key mediator of IL-1β production via the NOD-, LRR-, and pyrin domain-containing protein 3 (NLRP3) inflammasome (*deTorre-Minguela et al., 2016*). Lipopolysaccharide (LPS) treatment alone of the peritoneal macrophages from control and EROS knockout mice resulted in the secretion of modest amounts of IL-1β. Robust induction of IL-1β secretion was observed upon treatment of the LPS-conditioned macrophages with ATP, and this was approximately fourfold lower in EROS-deficient cells (*Figure 6B*). The requirement of EROS for P2X7-mediated calcium uptake and IL-1β production was previously reported in an independent study (*Ryoden et al., 2020*). In our in vivo data following intraperitoneal LPS and ATP injection, peritoneal washings from EROS knockout mice showed much reduced IL-1β secretion compared with control (*Figure 6C*). Consequently, caspase-1 activation was highly attenuated in EROS-deficient macrophages following LPS/ATP treatment (*Figure 6D*) compared to control macrophages. LPS treatment, either in vitro or in vivo, resulted in comparable levels of TNF-α,IL-10, IL-6, and KC/GRO secretion between control and EROS-deficient macrophages and levels of these cytokines were not altered significantly by ATP addition (*Figure 6—figure supplement 1A–F*).

P2X7 plays several important roles in CD4$^+$ T cell biology, including ADAM10/17-driven shedding of cell surface molecules such as CD62L and CD27 and externalisation of phosphatidylserine (PS), pore formation, and cell death (*Bartlett et al., 2014*; *Scheuplein et al., 2009*). Accordingly, flow cytometry analysis of CD62L level following ATP treatment showed reduced staining in control CD4$^+$ T cells with 10% of cells positive for CD62L-APC compared to 70% of positive cells pre-ATP addition (*Figure 6E*). In EROS-deficient CD4$^+$ T cells, this ATP-driven CD62L shedding is significantly impaired (*Figure 6E*). Similar results were obtained when examining CD27 shedding (*Figure 6F*). Nicotinamide adenine dinucleotide (NAD)-driven shedding of CD62L and CD27 was also significantly decreased in EROS-deficient CD4$^+$ T cells albeit to a lesser extent than with ATP treatment (*Figure 6—figure supplement 1G and H*). PS exposure and cell death, assessed by annexin V and propidium iodide staining, were reduced in EROS-deficient CD4$^+$ T cells following ATP treatment (*Figure 6G*) consistent with published findings (*Ryoden et al., 2020*).

We speculated that there might be further in vivo consequences to the loss of both P2X7 and phagocyte NADPH oxidase activity in EROS knockout mice and noted that blockade of the P2X7 receptor has been associated with improved outcomes in influenza A virus (IAV) infection (*Snelgrove et al., 2006*; *Leyva-Grado et al., 2017*; *Rosli et al., 2019*). We intranasally injected control and EROS knockout mice with IAV and found that EROS knockout mice were protected from infection with a 50–60% reduction in mortality (*Figure 6H*).

These results identify a central role of EROS in modulating purinergic signalling in vivo through targeting P2X7.

## Discussion

Cytochrome b558 is essential for defence against infection and a key modulator of inflammatory pathways. gp91*phox* exists as a co-dependent heterodimer with p22*phox* (*Segal, 1987*). gp91*phox* is translated as a 58 kDa immature protein which undergoes successive post-translational modifications, leading to the association with p22*phox* and final maturation to a functional 91 kDa protein (*DeLeo et al., 2000*). The 58 kDa form is inherently much more unstable and prone to proteasomal degradation than the mature 91 kDa form, as we also observed in this study. Recently, we showed that another protein, EROS, was essential for the expression of both gp91*phox* and p22*phox*. EROS deficiency led to a lack of gp91*phox* and p22*phox* (*Thomas et al., 2017b*; *Thomas et al., 2018b*), raising the question of how it fits into the canonical model above. Our study offers several new insights compared to previous work.

We show, by two different techniques, that EROS binds gp91*phox* directly and is needed very early in gp91*phox* biogenesis, in contrast to p22*phox*, preferentially stabilising the gp91*phox* 58 kDa precursor before heme incorporation takes place. Much work has been done to characterise the latter stages of gp91*phox* maturation (*Porter et al., 1996*; *Yu et al., 1997*; *Yu et al., 1999*; *DeLeo et al., 2000*; *Beaumel et al., 2014*). This study offers unique insight into the early stages of the protein's biosynthesis, on which all subsequent events depend. These early stages involve the OST-complex, which is required for the initial glycosylation events, folding and insertion into the ER membrane

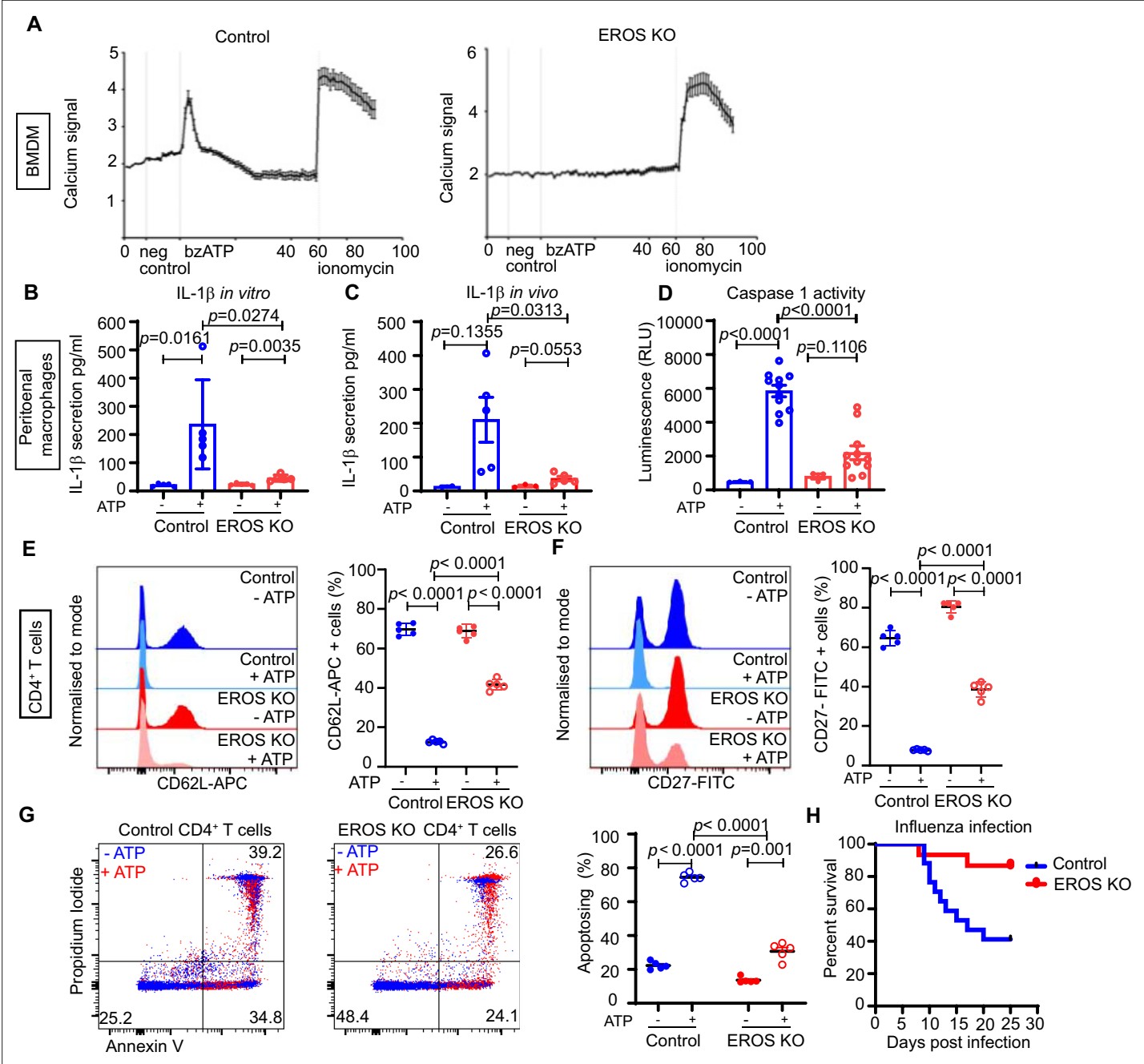

**Figure 6.** Functional consequences of low P2X7 expression in EROS-deficient cells. (**A**) Calcium release tested by Rhod-3-AM calcium imaging assay, in control or EROS-deficient bone marrow-derived macrophages (BMDM) in response to bzATP treatment (with ionomycin as positive control); data is representative of three independent experiments. (**B**) IL-1β secretion following lipopolysaccharide (LPS) priming of peritoneal macrophages from control or EROS knockout (KO) mice before and after treatment with ATP. (**C**) Secretion of IL-1β following in vivo administration of LPS and then ATP to control or EROS KO mice (n = 3–5 biological replicates). (**D**) Decreased caspase-1 activity, detected through luminescence production (see 'Methods'), in peritoneal macrophages from EROS KO mice compared to control mice following LPS priming and ATP treatment (n = 11 biological replicates). (**E, F**) Representative flow cytometry histogram of surface ligand CD62L (**E**) and CD27 (**F**) expression in CD4+ T cells isolated from EROS KO or control mice and treated with ATP (n = 5 biological replicates). Percentage of CD4+ T cells positive for CD62L or CD27 for each condition is shown on the left panel graphs. (**G**) Reduced phosphatidyl serine exposure and cell death in EROS-deficient CD4+ T cells compared to control CD4+ T cells following ATP treatment as analysed by flow cytometry staining with propidium iodide and annexin V. Left panel graph shows the percentage of CD4+ T cells undergoing apoptosis in each condition (representative of n = 5 biological replicates). (**H**) Control or EROS KO mice were infected intranasally with $3.10^3$ PFU of A/X31 influenza. Signs of illness were monitored daily (n = 17 control, 15 EROS knockout biological replicates). p-Value was determined using unpaired Student's t-test; error bars indicate SEM. See also *Figure 6—figure supplement 1*.

*Figure 6 continued on next page*

*Figure 6 continued*

The online version of this article includes the following figure supplement(s) for figure 6:

**Figure supplement 1.** Cytokine profile and surface ligand expression in EROS-deficient cell subsets.

(*Gemmer and Förster, 2020*). The OST complex adds a preformed oligosaccharide to an asparagine residue on a newly synthesised protein (*Cherepanova et al., 2016*). Murine gp91*phox* has one asparagine residue while human gp91*phox* has three asparagine residues that are glycosylated (*Harper et al., 1985*; *Wallach and Segal, 1997*). Using PNGaseF and EndoH-treated lysates as a reference, we demonstrated that EROS immunoprecipitated with partially glycosylated gp91*phox*. Blockade of OST-mediated N-glycosylation, through chemical inhibition or use of cells lacking the catalytic subunit STT3A, impacts on the gp91*phox* glycosylation pattern upon co-transfection with EROS in HEK293 cells but does not reduce the abundance of gp91*phox* per se as does EROS deficiency. Hence, we propose that EROS works post-translationally, stabilising gp91*phox* so that the high-mannose precursor can be synthesised and undergo further maturation.

We find that EROS can increase the abundance of NOX1, gp91*phox* (NOX2), and NOX4 in co-transfection assays and that it can directly bind NOX2, NOX3, and NOX4 in Yeast 2 Hybrid assays. This contrasts with its inability to augment the expression of NOX5 or p22*phox* and suggests that EROS likely binds a conserved motif in those NOX proteins that depend on p22*phox* for stability. We also present an EROS-deficient iPS-derived human macrophage line. This recapitulates the effects seen in cell lines and patients for gp91*phox*-p22*phox* but demonstrates for the first time in a primary human cell that EROS deficiency leads to P2X7 deficiency and that this is likely an important part of the human syndrome.

We place the role of EROS in context by using a granular mass spectrometry detecting up to 8000 proteins in different cell types to clearly define the range of proteins that EROS can regulate in context of the entire proteome. We used both whole-cell and plasma membrane to show just how specific EROS is in its physiological effects. In addition to our work on human iPS-derived macrophages, we make new insights into the biology of the control of P2X7 expression by EROS. The effects of EROS deficiency on P2X7 expression were noted in our original mass spectrometry analysis and in a recent screen of factors that influence ATP driven phosphatidylserine excursion (*Ryoden et al., 2020*). In this work, we show that EROS knockout mice have P2X7 deficiency in eight primary cell types of the murine immune system. Several splice variants exist in different tissues, and it is important to clarify whether EROS has cell type-specific effects on P2X7 expression. Notably in our T cell experiments, EROS-driven P2X7 deficiency affects shedding of CD27 and CD62L and ATP-driven cell death. Moreover, we show that the effects on P2X7 are a class effect with similar, but more modest, effects on P2X1 expression. We demonstrate that the mechanism by which this occurs is a direct effect on P2X7 and, by extension, is likely to be a direct effect on P2X1. Significantly, EROS overexpression drives increased abundance of P2X7 and P2X1, suggesting that EROS might act as a physiological regulator of purinergic signalling.

A particular strength of our study is that we show marked in vivo sequelae of the lack of P2X7. EROS deficiency leads to profound susceptibility to bacterial infection but protects mice from infection with influenza A. This is likely to reflect the fact that mice that are (i) deficient in gp91*phox,* (ii) deficient in P2X7, and (iii) treated with P2X7 inhibitors have improved outcomes following infection with influenza A and raises intriguing questions about the physiological role of EROS. Snelgrove et al. showed that gp91*phox* deficiency improved outcomes in influenza A. gp91*phox* knockout mice exhibited a reduced influenza titre in the lung parenchyma. Inflammatory infiltrate into the lung parenchyma was markedly reduced and lung function significantly improved (*Snelgrove et al., 2006*). To et al. then demonstrated that the phagocyte NADPH oxidase is activated by single-stranded RNA and DNA viruses in endocytic compartments. This causes endosomal hydrogen peroxide generation, which suppresses antiviral and humoral signalling networks via modification of a highly conserved cysteine residue (Cys98) on Toll-like receptor-7. In this study, targeted inhibition of endosomal ROS production using cholestanol-conjugated gp91dsTAT (Cgp91ds-TAT) abrogates IAV pathogenicity (*To et al., 2017*). This group went on to explore infection with a more pathogenic influenza A strain, PR8, using the same specific inhibitor. Cgp91ds-TAT reduced airway inflammation, including neutrophil influx and alveolitis and enhanced the clearance of lung viral mRNA following PR8 infection (*To et al., 2019*). This group has also shown that NOX1 (*Selemidis et al., 2013*) and NOX4 (*Hendricks et al.,*

*2022*) can drive pathogenic inflammation in influenza A, emphasising the importance of clarifying the roles of EROS in the control of the expression of these proteins.

In studies on P2X7, Rosli et al. found that mice infected with $10^5$ PFU of influenza A HKx31 had improved outcomes if they were treated with a P2X7 inhibitor at day 3 post infection and every 2 days thereafter. Survival was also improved even if the inhibitor is given on day 7 post infection following a lethal dose of the mouse-adapted PR8. This was associated with reduced cellular infiltration and pro-inflammatory cytokine secretion in bronchoalveolar lavage fluid, but viral titres were not measured (*Rosli et al., 2019*). Leyva-Grado et al. examined influenza A infection in P2X7 knockout mice. They infected mice with both influenza A/Puerto Rico/08/1934 virus and influenza A/Netherlands/604/2009 H1N1pdm virus. They showed that P2X7 receptor deficiency led to improved survival after infection with both viruses with less weight loss (*Leyva-Grado et al., 2017*). Production of proinflammatory cytokines and chemokines was impaired, and there were fewer cellular hallmarks of severe infection such as infiltration of neutrophils and depletion of CD11b$^+$ macrophages. It is worth noting that the P2X7 knockout strain used in this study was the Pfizer strain in which some splice variants of P2X7 are still expressed (*Bartlett et al., 2014*). Hence, the dual loss of the phagocyte NADPH oxidase and P2X7 in EROS knockout mice likely confers protection from IAV infection. By reducing the expression of both NOX2 and P2X7, EROS regulates two pathways that may be detrimental in influenza A and we speculate that EROS may physiologically act as a rheostat controlling certain types of immune response.

P2X7 is crucial for ATP-driven activation of the NLRP3 inflammasome and has an emerging role in T cell biology, especially in resident immune cells (*Stark, 2018*). Of note, P2X7 ligation by ATP is also a potent driver of ROS formation, the other major process that is profoundly affected by EROS deficiency (*Pfeiffer et al., 2007*). One role of EROS might be to modulate ATP-driven danger signalling including ROS and NLRP3 activation through controlling the biosynthesis of NOX2 and P2X7. Both ROS and NLRP3 activation are clearly essential, but the extent and duration of responses must be controlled to limit tissue damage. Therefore, one might consider EROS as a rheostat for these key linked immune processes.

In conclusion, we have elucidated the mechanism by which EROS controls the abundance of gp91*phox* and purinergic receptors and defined EROS as a highly selective molecular chaperone that can shape immune responses through targeting specific proteins. As well as advancing our understanding of the physiology of highly conserved and non-redundant proteins, this work has implications for translational medicine, including gene therapy and possible therapeutic blockade of EROS to limit viral immunopathology.

## Methods
### Animal use
EROS/*Cybc1*$^{-/-}$ mice (KOMP) and gp91*phox*/*Cybb*$^{-/-}$ mice were previously described (*Thomas et al., 2017b*). The care and use of all mice were in accordance with UK Home Office regulations (UK Animals Scientific Procedures Act 1986). The mice were maintained in specific pathogen-free conditions and matched by age and sex within experiments. For the mouse infection, groups of >15 isofluorane-anaesthetised mice of wild-type or EROS knockout genotype were intranasally inoculated with $3.10^3$ PFU of A/X-31 influenza in 50 µL of sterile PBS. Their weight was recorded daily, and they were monitored for signs of illness. Mice exceeding 25% total weight loss were killed in accordance with UK Home Office guidelines.

### DNA constructs
Untagged pEGBacMam-p22*phox* was amplified from OHu21427 (GenScript) by PCR and subcloned in pEGBacMam vector, a kind gift from Eric Gouaux (Oregon Health & Science University; *Goehring et al., 2014*) using In-Fusion seamless DNA cloning (Takara Bio). PCR was used to generate the pEGBacMam-Strep-FLAG-Strep-EROS construct by amplifying the human EROS/*CYBC1* gene and contextually inserting the Strep-FLAG-Strep tag at its 5'. Human NOX1 was synthesised by Genewiz with a Strep II-FLAG tag at the N-terminus and cloned into the pEGBacMam vector using In-Fusion seamless DNA cloning (Takara Bio). The cDNAs encoding for human NOX5 (isoform b) were

synthesised by GeneArt and subcloned into the pNGFP-EU vector, a kind gift from Eric Gouaux (*Kawate and Gouaux, 2006*).

## Cell line culture

HEK293T, HEK293, control HEK293, *STT3A*$^{-/-}$ HEK293, *STT3B*$^{-/-}$ HEK293, COS-7, NIH/3T3, and RAW 264.7 EROS FLAG-tagged cells were maintained in DMEM medium (21969035, Thermo Fisher) containing 10% FBS (F7524, Sigma-Aldrich) and 100 U/mL of Penicillin-Streptomycin-Glutamine (10378016, Thermo Fisher). PLB985 (ACC 139, DSMZ), PLB-clone 14 (*Thomas et al., 2018b*), PLB- clone 20 (*Thomas et al., 2018b*), and PLB985 overexpressing EROS- GFP lentivirus construct were cultured in complete RPMI medium consisting of RPMI 1640 (31870025, Thermo Fisher), 10% FBS (F7524, Sigma-Aldrich), 2 mM GlutaMAX (35050061, Thermo Fisher), 1 mM sodium pyruvate (11360070, Thermo Fisher); 0.5 mg/mL Penicillin-Streptomycin-Glutamine (10378016, Thermo Fisher), and 20 mM HEPES (H3537, Sigma-Aldrich). HEK293-F cells were grown in suspension using FreeStyle medium (Invitrogen). All cell lines were tested and confirmed mycoplasma-free using the protocol for Myco-Alert Mycoplasma Detection Kit (LT07-218, Lonza).

## Primary cell culture

To generate BMDM, bone marrow cells harvested from femurs and tibias of 6–10-week-old control or EROS knockout mice were grown in complete RPMI medium supplemented with 100 ng/mL of murine M-CSF (PeproTech) for 3 days. At day 3, 10 mL of the same medium was added to the culture and differentiated macrophages were collected at day 7 for mass spectrometry or Western blot analysis. To isolate mouse CD4 lymphocytes, mouse spleens were homogenised by manual disruption and subjected to positive selection using CD4 L3T4 magnetic beads (Miltenyi Biotec) according to the manufacturer's protocol. Cells were counted and resuspended at a concentration of $2 \times 10^6$ cells per mL in complete RPMI medium for subsequent analysis.

Control or EROS-deficient human iPS were generated by CRISPR Cas 9-targeted deletion of 46 base pairs in exon 5 of the *CYBC1* sequence (*Yeung et al., 2017*). Human macrophages were obtained from the differentiation of the human control or EROS-deficient iPS line following previously established method (*van Wilgenburg et al., 2013*; *Thomas et al., 2017b*). Macrophages were cultured in complete RPMI in the presence of human M-CSF (PeproTech) and were used at day 7 post-differentiation.

## Transient expression

Cells were transfected at 60–80% confluency with 1.6–2 μg of the following constructs: mouse gp91*phox*, mouse GFP-tagged gp91*phox*, mouse EROS, human EROS, human gp91*phox*, human mRFP-tagged gp91*phox*, human p22*phox*, human NOX4, mouse GFP-tagged P2X7, human P2X1, and human P2X4; using Lipofectamine RNAiMAX (Thermo Fisher) reagent for HEK293T, HEK293, and COS-7 cells and Lipofectamine 2000 (Thermo Fisher) reagent for NIH3T3 cells, following the manufacturer's recommendation. HEK293-F were transfected with polyethylenimine (Polyscience Europe GmbH) at a ratio of 1:3 DNA:polyethylenimine (*Tom et al., 2008*) using 0.5 μg of the indicated constructs. Equivalent amounts of the corresponding empty vector were used as negative control. Cells were harvested and analysed 48–72 hr post-transfection.

## Protein stability and drug treatment

For cycloheximide experiment, HEK293T cells (transfected with a gp91*phox* construct alone or in combination with an EROS construct) and PLB985 cells (parental and expressing lentivirus EROS-GFP) were treated with 10 μg/mL of cycloheximide (Sigma) and harvested at 2 hr, 4 hr, 6 hr, and 8 hr post-treatment and subjected to Western blot. Non-treated cells were used as control. For heme synthesis inhibition, 10 μg/mL succinyl acetone (Merck) was added to the cell culture 4 hr after transfection. For glycosylation inhibition, NGI-1 compound (Tocris) or tunicamycin (Sigma) was added 3 hr post-transfection of HEK293 cells and left until harvesting (48–72 hr later).

## Immunoprecipitation and SEC

Strep tagged-EROS was co-transfected in HEK293-F cells with GFP-gp91*phox* and p22*phox*. Then, 48 hr post-transfection cells were harvested and cell membranes were prepared as described in

*Ceccon et al., 2017*. Membranes were solubilised with 1% LMNG in 50 mM HEPES pH 7.5, 100 mM NaCl, 20% (v/v) glycerol, and passed through Streptactin-resin. Bound proteins were eluted from the column with 5 mM desthiobiotin in 50 mM HEPES pH 7.5, 100 mM NaCl, 5% (v/v) glycerol, and then subjected to SEC using a Superose 6 30/150 column.

Lysates from $3.10^7$ to $4.10^7$ of cells were subjected to immunoprecipitation according to the manufacturer's protocol for the Sure Beads Protein G Magnetic Beads (161-4023, Bio-Rad) using a rat anti-FLAG antibody (BioLegend), the mouse anti-gp91*phox* antibody (Santa Cruz), and an IgG2aK isotype control antibody (14-4321-82, Invitrogen). Eluates from FLAG, gp91*phox,* and IgG control beads were analysed by Western blotting. Where indicated, lysates were treated with PNGase F or EndoH (New England Biolabs) following the manufacturer's recommendation.

## Western blot analysis

Cells were lysed at a concentration of $2.10^7$ cells per mL in Pierce RIPA buffer (89900, Thermo Fisher) containing cOmplete Protease Inhibitor Cocktail (11697498001, Sigma Aldrich) and Halt Protease and Phosphatase inhibitor (78440, Thermo Fisher). Protein concentration was determined by BCA assay (23225, Thermo Fisher) according to the manufacturer's instructions. 15–20 µg of protein, mixed with 4X NuPAGE sample buffer (NP0008, Thermo Fisher), were resolved on a NuPAGE 4–12% Bis-Tris gel (NP0336BOX; NP0335BOX; Thermo Fisher) under reducing conditions in MOPS buffer (mops-sds1000, Formedium), transferred to a nitrocellulose membrane (GE10600003, Sigma-Aldrich) in NuPAGE transfer buffer (NP00061; Thermo Fisher), and probed with one of the following primary antibodies: mouse anti-gp91*phox* (Santa Cruz Biotechnology), rabbit anti-C17ORF62/EROS (Atlas), rabbit anti-p22*phox* (Santa Cruz Biotechnology), mouse anti-p22*phox* (Santa Cruz Biotechnology), rabbit anti-P2X1 (Alomone), rabbit anti-P2X4 (Alomone), rabbit anti-P2X7 (Alomone; Atlas), rabbit anti-P2X7 (Atlas Antibodies), rabbit anti-vinculin (Cell Signalling Technology), rabbit anti-actin (Abcam), mouse anti-α-tubulin (Abcam), mouse anti-FLAG-M2 (Sigma), a homemade rat anti-NOX1 antibody given by Dr. Misaki Matsumoto, and a rabbit anti-NOX4 antibody from Prof. Ajay Shah (King's College London, UK). Secondary antibodies used were anti-rabbit IgG-horseradish peroxidase (7074S, Cell Signaling Technology, dilution 1:10,000), anti-rat IgG-horseradish peroxidase (62-9520, Thermo Fisher, dilution 1:5000), and anti-mouse IgG-horseradish peroxidase (7076S, Cell Signaling Technology, dilution 1:10,000). Blots were developed using one of the following: ECL (32106, Thermo Fisher), SuperSignal West Pico PLUS (34577, Thermo Fisher), or SuperSignal West Femto (34095, Thermo Fisher) reagents, and chemiluminescence was recorded on a ChemiDoc Touch imager (Bio-Rad).

## NanoBiT assay for protein–protein interaction

The assay was performed following the manufacturer's protocol for the MCS Starter System (N2014, Promega). The constructs encoding EROS/*CYBC1* or gp91*phox*/*CYBB* fused to the reporter subunits were custom-made by Promega and consist of the following: SmBiT-*CYBC1* (TK) vector (CS1603B224), *CYBC1*-SmBiT (TK) vector (CS1603B225), LgBiT-*CYBC1* (TK) vector, (CS1603B226), *CYBC1*-LgBiT (TK) vector (CS1603B227), SmBiT-*CYBB* (TK) vector (CS1603B228), *CYBB*-SmBiT (TK) vector (CS1603B229), LgBiT-*CYBB* (TK) vector (CS1603B230), and *CYBB*-LgBiT (TK) vector (CS1603B231). Promega also generated a LgBiT-*CYBC1*-*CYBB*-SmBiT BiBiT vector (CS1603B292) and a LgBiT-*CYBC1* Bi-ready Vector (CS1603B290), where fusions proteins are expressed from a single bidirectional CMV promoter. NanoBiT Bi-Directional Vector Systems allow controlling transfection efficiency of separate constructs. HEK293 were plated at a concentration of $2.10^4$ cells per well in a 96-well plate (3917, Corning). The following day, cells were transfected either with 55 ng/well of the indicated vector combinations (1:1 ratio of interacting pairs), the Bi-BiT ready Vectors, or with the NanoBIT-negative and -positive control provided (not shown in graphs) using FugeneHD reagent (E2311, Promega) at a lipid to DNA ratio of 3:1. Then, 20 hr post-transfection, growth medium was exchanged with OPTI-MEM (51985034, Thermo Fisher) containing 2% FBS (7534, Sigma Aldrich) and 20 mM HEPES (Thermo Fisher). The plate was left to equilibrate 10 min at room temperature before addition of the freshly reconstituted Nano-Glo Live cell substrate. Luminescence was subsequently measured on a FLUOstar Omega plate reader (BMG, Labtech) using the following settings: plate mode, number of cycles: 8; cycle interval: 900 s, gain adjustment 70% of the target (positive control), orbital averaging of the well: 3; orbital shaking 300 rpm for 15 s before the first measurement.

## Yeast 2 Hybrid experiment

gp91*phox* and EROS analysis was performed by Hybrigenics Services (91000 Evry, France). The coding sequences of human EROS/*CYBC1* (NM_001033046.3) and human gp91*phox*/*CYBB* (NM_000397.3) were PCR-amplified and cloned in frame with the Gal4 DNA binding domain (DBD) into plasmid pB66 (**Fromont-Racine et al., 1997**) as a C-terminal fusion to Gal4 (Gal4-DBD-bait fusion) and with the Gal4 Activation Domain (AD) into plasmid pP7 (AD-prey fusion). The diploid yeast cells were obtained using a mating protocol with both yeast strains (**Fromont-Racine et al., 1997**), based on the HIS3 reporter gene (growth assay without histidine). As negative controls, the bait plasmids were tested in the presence of empty prey vector and the prey plasmids were tested with the empty bait vector. The interaction between SMAD and SMURF is used as positive control (**Colland et al., 2004**). Controls and interactions were tested in the form of streaks of three independent yeast clones for each control and interaction on DO-2 and DO-3 selective media. The DO-2 selective medium lacking tryptophan and leucine was used as a growth control and to verify the presence of the bait and prey plasmids. The DO-3 selective medium without tryptophan, leucine, and histidine selects the interaction between bait and prey. For specific interactions, the selection pressure was increased using 0.5 mM 3-aminotriazol (3-AT). Yeast 2 Hybrid analysis of NOX family of proteins was performed as previously reported (**Luck et al., 2020**).

## Cytokine secretion

For in vivo experiments, EROS knockout mice were injected with 250 ng/mL of LPS (Invivogen). Then 2 hr later, 30 mM of ATP or PBS (for control) was injected intraperitoneally. Mice were culled 2 hr later, and the peritoneal cavity washed. Supernatants and serum were analysed for the presence of the cytokines indicated in the figures using multiplex kits from Meso Scale Discovery (MD, USA) and were conducted at the Core Biochemical Assay Laboratory (Cambridge University Hospital, UK).

## Caspase-1 activity measurement

Peritoneal macrophages from control or EROS knockout mice were primed overnight with LPS (100 ng/mL, Invivogen) and treated for 2 hr with ATP (2.5 mM, Invivogen). Supernatants were harvested and subjected to Caspase-Glo 1 inflammasome assay following the manufacturer's protocol (G9951, Promega). Luminescence from caspase-1 activity was detected with a FLUOstar Omega plate reader (BMG, Labtech).

## Immunofluorescence

Cells expressing human pEGFP-N2-STT3A, human pmRFP-N2-EROS/C17ORF62, and human pEGFP-N2-Lap2β constructs (generously given by Prof. Eric Schirmer, University of Edinburgh, UK) were fixed 15 min in 4% paraformaldehyde (15710S, Electron Microscopy), permeabilised with 0.1% Triton X-100 (Sigma), and quenched in 100 mm glycine (Sigma) for 15 min. To reduce background, a blocking solution consisting of 5% goat serum (Sigma), 1% BSA (Sigma) in PBS (Sigma) was applied for 1 hr prior to staining with the indicated primary antibody diluted in PBS solution with 0.5% goat serum (Sigma), 0.5% BSA (Sigma) for 1h30 at room temperature. Following extensive washes, cells were stained for 45 min at room temperature with the appropriate secondary antibodies from Life Technologies: goat anti-mouse IgG–Alexa 488 conjugated (A11029), goat anti-rabbit IgG–Alexa 555 conjugated (A21429), and/or goat anti-rabbit IgG–Alexa 647 (A21245). Nucleus was labelled with Hoechst 33342 (H3570, Thermo Fisher) before coverslips were mounted in Prolong Diamond (P36961, Life Technologies).

Images were acquired on a Zeiss LSM 780 system, equipped with the following lasers: diode 405 nm, argon multiline 458/488/514 nm, HeNe 543 nm, HeNe 594 nm, and HeNe 633 nm and using a Plan Apochromat ×63/1.4 oil objective. Image processing and analysis of the intensity profile were done on Fiji using just ImageJ software (version 1.53c) using the 'plot profile' command.

## Calcium flux assay

Calcium imaging was performed on EROS-deficient and control cells using the Rhod-3-AM calcium imaging kit (Invitrogen) following the manufacturer's protocol. Images were acquired with an LSM710 laser scanning META confocal microscope (Carl Zeiss) using a ×20 objective and maximum pinhole aperture of 600 μm. Two-line averages were performed for each frame with images taken every 4 s.

Then, 20 µL of media was used as a negative control. At t = 140 s, 100 µM bzATP was added to the cells. Then, 100 ng/mL ionomycin was used as a positive control at the end of the experiment to confirm correct loading of cells. Image analysis of fluorescence intensity in response to addition of negative control, bzATP, and positive control across the time course was performed with the Volocity 3D Image Analysis Software, collecting data for all cells that did not spontaneously fluoresce and had an ionomycin response.

## Flow cytometry

Single-cell suspensions of spleens were prepared by mechanical disruption in FACS buffer (D-PBS with 2 mM EDTA, 0.5% FBS, and 0.09% sodium azide) with a 100 µm smart strainer (Miltenyi Biotec). Then, 10% of the spleen was subjected to red blood cell lysis (eBioscience), washed, and blocked with 1 µg of Mouse BD FC Block for 10 min at 4°C prior to addition of multicolour antibody cocktails. After incubation for 30 min, DAPI was added (0.2 µg/mL) and the samples washed prior to acquisition on a LSRFortessa (BD Biosciences) that was standardised using BD Cytometer Setup and Tracking beads and software. Compensation was determined using Ultracomp eBeads (eBioscience). Data acquisition was controlled with BD FACSDiva v8 software. The antibody cocktails included anti-mouse antibodies from Miltenyi Biotec: CD44-FITC (clone IM7.8.1), CD62L-PerCP-Vio700 (clone REA828), CD25-PE-Vio770 (clone 7D4), NK1.1-APC (clone PK136), TCRγδ-Vioblue (clone GL3), and CD4-Viogreen (clone GK1.5). Mouse anti-CD8α-APC-H7 (clone 53-6.7) and mouse TCRβ-BV711 (clone H57.597) were from BD Biosciences. Mouse anti-P2X7-PE (clone 1F11) and mouse anti-CD45-AF700 (clone 30-F11) were from BioLegend.

All samples were analysed using FlowJo 10.7 in a blinded manner. Doublets were excluded via FSC-A vs. FSC-H and SSC-H vs. SSC-W and dead cells excluded by DAPI vs. FSC-A. A plot of CD45 against time was used to check the stability of sample acquisition and leukocytes identified by a CD45 vs. SSC-A plot. CD4 T cells are TCRβ+NK1.1-CD4+, CD8 T cells are TCRβ+NK1.1-CD8+, NK cells are TCRβ-NK1.1+, NKT cells are TCRβ+NK1.1+, γδT cells are TCRγδ+TCRβ-, and B cells were TCRβ-NK1.1-TCRγδ-SSC low. A gate for P2X7 was set on each population based on a fluorescence minus one control.

For shedding analysis, 100 µl of CD4 cells were treated with either 29 µM NAD (Sigma) or 2.5 mM ATP (Invivogen) for 30 min at 37°C, washed, resuspended in FACS buffer (PBS, 2% FBS), and co-stained with anti-CD4 (GK1.5, BioLegend), anti-CD62L (MEL-14, BioLegend), and anti-CD27 (LG.3A10, BioLegend) for 10 min at room temperature. Unbound antibodies were removed by washing cells in FACS buffer, and cells were kept in the same buffer for the acquisition.

For cell death analysis, NAD or ATP-treated CD4 cells (as described above) were resuspended in annexin-binding buffer (Thermo Fisher) and incubated with 1 µL 100 µg/mL propidium iodide (PI; Thermo Fisher) and 1/50 annexin V antibody (Thermo Fisher) for 10 min at room temperature. Cells were washed and resuspended in 1× annexin-binding buffer for the acquisition. Samples were acquired on an LSR X20 flow cytometer (BD).

## EROS affinity purification-mass spectrometry

EROS-FLAG affinity purifications were carried out as described previously (*Thomas et al., 2017b*). Bound proteins were eluted by incubating the beads with 200 µg/mL 3xFLAG peptide (Sigma-Aldrich) in IPP150 containing 0.02% NP-40. The eluates were concentrated in Vivaspin 500 PES centrifugal filters (Vivascience), reduced with 5 mM TCEP (Sigma-Aldrich), and alkylated with 10 mM iodoacetamide prior to sample fractionation by polyacrylamide gel electrophoresis with Novex NuPAGE Bis-Tris 4–12% gels (Life Technologies). Gels were stained with colloidal Coomassie (Sigma), and whole-gel lanes were excised into 12 bands and processed for mass spectrometry analysis as previously described (*Pardo et al., 2010*).

Peptides were reconstituted in 0.1% formic acid and injected for online LC-MS/MS analysis on an LTQ FT Ultra hybrid mass spectrometer coupled with an UltiMate 3000 RSLCnano UPLC system. Peptides were separated on a PepMap C18 column (75 µm i.d. × 250 mm, 2 µm) over a linear gradient of 4–33.6% CH3CN/0.1% formic acid in 60 min at a flow rate at 300 nL/min. MS analysis used standard data-dependent acquisition mode to fragment the top 5 multiply charged ions with intensity above 1000. Briefly, the FT full MS survey scan was m/z 380–1800 with resolution 100,000 at m/z 400, AGC set at 1e6, and 500 ms maximum injection time. The MS/MS fragmented in ion trap was set at 35%

collision energy, with AGC at 1e4 and 250 ms maximum injection time, and isolation width at 2.0 Da. The dynamic exclusion was set 45 s with ±20 ppm exclusion window.

Raw files were processed with Proteome Discover 2.3 (Thermo Fisher Scientific). Database searches were performed with Sequest HT against the mouse UniProt database (v. August 2019) and cRAP contaminant database. The search parameters were trypsin, maximum of two missed cleavages, 10 ppm mass tolerance for MS, 0.5 Da tolerance for MS/MS, with variable modifications of carbamidomethyl (C), N-acetylation (protein N-term), deamidation (NQ), oxidation (M), formylation (N-term), and Gln->pyro-Glu (N-term Q). Database search results were refined through processing with Percolator (FDR <1%). Protein identification required at least one high-confidence peptide. External contaminants (keratins, albumin, casein, immunoglobulins) were removed before further analysis. SAINTexpress was used to score interaction specificity (*Teo et al., 2014*). Proteins with SAINT probability score >0.9 were deemed high-confidence-specific interactors (FDR <1%). Proteins with SP score >0.74 represent medium-confidence interactors (FDR <5%).

For visualisation, the protein interaction network was generated with STRING (minimum required interaction score 0.4) using interactions derived from text mining, experiments, and databases. GO term enrichment analysis was performed with STRING.

## Full proteome quantitative analysis by TMT-mass spectrometry

Cell pellets were lysed in 1% sodium deoxylate/10% isopropanol/50 mM NaCl/100 mM tetraethylammonium bromide (TEAB, Sigma) with Halt Protease and Phosphatase Inhibitor Cocktail (1×, Thermo Scientific). Lysates were sonicated and heated at 90°C for 5 min. Then, 100 µg of proteins per sample were reduced with TCEP, alkylated with iodoacetamide, and digested with 3 µg of trypsin (Pierce MS grade, Thermo) prior to labelling with TMT10plex according to the manufacturer's instructions. Samples were pooled, acidified, and then centrifuged to remove precipitated deoxycholic acid. The supernatant was dried, resuspended in 0.1% $NH_4OH$, and fractionated on an XBridge BEH C18 column (2.1 mm i.d. × 150 mm, Waters) with an initial 5 min loading then linear gradient from 5% $CH_3CN$/0.1% $NH_4OH$ (pH 10)–35% $CH_3CN$ /0.1% $NH_4OH$ for 30 min, then to 80% $CH_3CN$/0.1% $NH_4OH$ for 5 min and stayed for another 5 min. The flow rate was at 200 µL/min. Fractions were collected every 42 s from 7.8 min to 50 min and then concatenated to 28 fractions and dried.

Peptides were reconstituted in 0.1% formic acid and a 25% aliquot injected for online LC-MS/MS analysis on the Orbitrap Fusion Lumos hybrid mass spectrometer coupled to an UltiMate 3000 RSLC-nano UPLC system (Thermo Fisher). Samples were desalted on a PepMap C18 nano trap (100 µm i.d. × 20 mm, 100 Å, 5µ), then separated on a PepMap C18 column (75 µm i.d. × 500 mm, 2 µm) over a linear gradient of 5.6–30.4% $CH_3CN$/0.1% formic acid in 90 min at a flow rate at 300 nL/min. The MS acquisition used MS3 level quantification with Synchronous Precursor Selection (SPS5) with the Top Speed 3 s cycle time. Briefly, the Orbitrap full MS survey scan was m/z 375–1500 with the resolution 120,000 at m/z 200, with AGC set at 4e5 and 50 ms maximum injection time. Multiply charged ions (z = 2–5) with intensity threshold at 5000 were fragmented in ion trap at 35% collision energy, with AGC at 1e4 and 50 ms maximum injection time, and isolation width at 0.7 Da in quadrupole. The top 5 MS2 fragment ions were SPS selected with the isolation width at 0.7 Da, fragmented in HCD at 65% NCE, and detected in the Orbitrap. The resolution was set at 50,000, and the AGC at 1e5 with maximum injection time at 86 ms. The dynamic exclusion was set 40 s with ±10 ppm exclusion window.

Raw files were processed with Proteome Discoverer 2.4 (Thermo Fisher) using Sequest HT. Spectra were searched against UniProt mouse database (April 2020) and cRAP contaminant database. Search parameters were trypsin with two maximum miss-cleavage sites, mass tolerances at 20 ppm for precursors and 0.5 Da for fragment ions, dynamic modifications of deamidated (N, Q), oxidation (M), and acetyl (protein N-terminus), static modifications of carbamidomethyl (C) and TMT6plex (peptide N-terminus and K). Peptides were validated by Percolator with q-value set at 0.01 (strict) and 0.05 (relaxed). The TMT10plex reporter ion quantifier included 20 ppm integration tolerance on the most confident centroid peak at the MS3 level. The co-isolation threshold was set at 100%. Only unique peptides, with average reported signal-to-noise ratio >3, were used for protein quantification, and the SPS mass matches threshold was set at 55%. Only master proteins were reported.

## Plasma membrane profiling analysis by tandem mass tagging-mass spectrometry

Cell surface proteins were labelled essentially as described previously (*Weekes et al., 2012*). Samples were then resuspended in 21 µL 100 mM TEAB pH 8.5 prior to labelling with tandem mass tagging (TMT) reagent (Thermo Fisher) following the manufacturer's protocol. After checking whether each sample was at least 98% TMT labelled, total reporter ion intensities were used to normalise the pooling of the remaining samples to a 1:1 ratio of total peptide content between samples. This final pool was brought up to a volume of 1 mL with 0.1% TFA. FA was added until the pH was <2. Samples were then cleaned up by SPE using a 50 mg tC18 SepPak cartridge (Waters). The cartridge was wetted with 1 mL 100% methanol followed by 1 mL ACN, equilibrated with 1 mL 0.1% TFA, and sample loaded slowly. Samples were passed twice over the cartridge. The cartridge was washed 3× with 1 mL 0.1% TFA before eluting sequentially with 250 µL 40% ACN, 70% ACN, and 80% ACN and dried in a vacuum centrifuge.

TMT-labelled samples were resuspended in 40 µL 200 mM ammonium formate pH 10 and transferred to a glass HPLC vial for basic pH reversed-phase fractionation (BpH-RP). BpH-RP fractionation was conducted on an UltiMate 3000 UHPLC system (Thermo Scientific) equipped with a 2.1 mm × 15 cm, 1.7 µm Kinetex EVO column (Phenomenex). Solvent A was 3% ACN, solvent B was 100% ACN, and solvent C was 200 mM ammonium formate (pH 10). Throughout the analysis, solvent C was kept at a constant 10%. The flow rate was 500 µL/min and UV was monitored at 280 nm. Samples were loaded in 90% A for 10 min before a gradient elution of 0–10% B over 10 min (curve 3), 10–34% B over 21 min (curve 5), 34–50% B over 5 min (curve 5), followed by a 10 min wash with 90% B. Then, 15 s (100 µL) fractions were collected throughout the run. Fractions containing peptide (as determined by A280) were recombined across the gradient to preserve orthogonality with online low pH RP separation. For example, fractions 1, 25, 49, 73, and 97 were combined and dried in a vacuum centrifuge and stored at –20°C until LC-MS analysis. Twelve fractions were generated in this manner.

For mass spectrometry, analysis was conducted on an Orbitrap Fusion instrument online with an UltiMate 3000 RSLCnano UHPLC system (Thermo Fisher). Samples were resuspended in 10 µL 5% DMSO/1% TFA, and all samples were injected. Trapping solvent was 0.1% TFA, analytical solvent A was 0.1% FA, and solvent B was ACN with 0.1% FA. Samples were loaded onto a trapping column (300 µm × 5 mm PepMap cartridge trap (Thermo Fisher)) at 10 µL/min for 5 min at 60°C. Samples were then separated on a 75 cm × 75 µm i.d. 2 µm particle size PepMap C18 column (Thermo Fisher) at 55°C. The gradient was 3–10% B over 10 min, 10–35% B over 155 min, 35–45% B over 9 min, followed by a wash at 95% B for 5 min and re-equilibration at 3% B. Eluted peptides were introduced by electrospray to the MS by applying 2.1 kV to a stainless steel emitter 5 cm × 30 µm (PepSep). Mass spectra were acquired using MS3 level quantification with SPS10 with the Top Speed 3 s cycle time. The Orbitrap full MS survey scan was m/z 400–1500 with the resolution at 120,000, AGC set at 125 and 50 ms maximum injection time. Multiple charged ions (z = 2–7) with intensity threshold at 5000 were fragmented in ion trap at 35% collision energy, with AGC at 80 and isolation width at 0.7 Da in quadrupole. The top MS3 fragment ions were SPS selected with the isolation width at 2 and were fragmented in HCD at 65% and detected in Orbitrap at 50,000 resolution. The AGC was set at 40 with maximum injection time at 120 ms, and the dynamic exclusion was set at 40 s with ±10 ppm exclusion window.

Data were processed with MASCOT (Matrix Science) and Proteome Discoverer, v2.2 (Thermo Fisher). Raw files were searched against the UniProt Mouse database, including common contaminants at a MS1 Tol of 10 ppm, MS2 Tol of 0.5 Da. Fixed modifications were TMT-labelled lysines and Peptide N-Termini and carbamidomethylated cysteines. Methionine oxidation was allowed as a variable modification. Mascot Percolator was used to control the PSM FDR, and an automated decoy database search was used to determine protein FDR. Proteins with either 'high' (FDR <0.01) or 'medium' (FDR <0.05) confidence were taken forward to further analysis. Proteins and their abundances were output to .csv, imported to R, and submitted to statistical analysis using LIMMA, a moderated *t*-test available through the Bioconductor package. LIMMA p-values were corrected for multiple hypothesis testing using the Benjamini–Hochberg method to generate an FDR (q-value).

## Data availability

The mass spectrometry proteomics data have been deposited to the ProteomeXchange Consortium via the PRIDE (*Perez-Riverol et al., 2019*) partner repository with the dataset identifier 'PXD024659.' The plasma membrane profiling mass spectrometry data has the identifier 'PXD025149'.

## Acknowledgements

DCT is funded by a Wellcome-Beit Prize Trust Clinical Research Career Development Fellowship and the Burman Fund, Imperial College London. EG and ERD are funded by the Wellcome Trust. This work was supported by the Italian Ministry of Education, University and Research (MIUR, 'Dipartimenti di Eccellenza Program 2018–2022—Dept. of Biology and Biotechnology L Spallanzani,' University of Pavia). We are grateful to Prof. Neil Bulleid for sharing the control, *STT3A*$^{-/-}$ and *STT3B*$^{-/-}$ HEK293 originally from Prof. Reid Gilmore. We thank Prof. Arthur Kaser, Prof. Eric Schirmer, Prof. Ramanujan Hedge, Prof. Marina Botto, Prof. Matthew Pickering, and Dr. Jacques Behmoaras for helpful discussion. We thank the Facility for Imaging by Light Microscopy (FILM) at Imperial College London (London, UK), partly supported by funding from the Wellcome Trust (grant 104931/Z/14/Z) and BBSRC (grant BB/L015129/1). We thank the LMS/NIHR Imperial Biomedical Research Centre Flow Cytometry Facility for the support. We also thank the Microscopy facility at the Cambridge Institute of Medical Research (Cambridge, UK).

## Additional information

### Funding

| Funder | Grant reference number | Author |
| --- | --- | --- |
| Wellcome Trust | 206617/A/17/Z | David C Thomas<br>Elizabeth R Dufficy<br>Emma Garside<br>Esme Nichols |

The funders had no role in study design, data collection and interpretation, or the decision to submit the work for publication. For the purpose of Open Access, the authors have applied a CC BY public copyright license to any Author Accepted Manuscript version arising from this submission.

### Author contributions

Lyra O Randzavola, Conceptualization, Investigation, Visualization, Writing - original draft, Writing - review and editing; Paige M Mortimer, Emma Garside, Elizabeth R Dufficy, Andrea Schejtman, Georgia Roumelioti, Charlotte Tolley, Cordelia Brandt, Katherine Harcourt, Mike Nahorski, James C Williamson, Shreehari Suresh, John M Sowerby, William M Rae, Anneliese Speak, Investigation; Lu Yu, Mercedes Pardo, Data curation, Investigation; Kerstin Spirohn, Validation, Methodology; Esme Nichols, Validation; Geoff Woods, Subhankar Mukhopadhyay, Jyoti Choudhary, Simon Clare, Giorgia Santilli, Kenneth GC Smith, Resources; Misaki Matsumoto, Celio XC Santos, Cher Shen Kiar, Michael A Calderwood, Resources, Methodology; Gordon J Dougan, Conceptualization; John Grainger, Conceptualization, Resources, Methodology; Paul J Lehner, Conceptualization, Resources; Alex Bateman, Data curation, Visualization, Methodology; Francesca Magnani, Conceptualization, Resources, Funding acquisition, Investigation; David C Thomas, Conceptualization, Funding acquisition, Investigation, Writing - original draft, Project administration

### Author ORCIDs

Lyra O Randzavola (ID) http://orcid.org/0000-0003-3463-1858
Paul J Lehner (ID) http://orcid.org/0000-0001-9383-1054
Alex Bateman (ID) http://orcid.org/0000-0002-6982-4660
Francesca Magnani (ID) http://orcid.org/0000-0003-0812-9397
David C Thomas (ID) http://orcid.org/0000-0002-9738-2329

### Ethics

The care and use of all mice were in accordance with UK Home Office regulations (UK Animals Scientific Procedures Act 1986).

### Decision letter and Author response

Decision letter https://doi.org/10.7554/eLife.76387.sa1
Author response https://doi.org/10.7554/eLife.76387.sa2

## Additional files

### Supplementary files

• Supplementary file 1. Primers used to generate induced pluripotent stem cells (iPS) knockout for EROS by CRISPR. (A) Sequences of the CRISPR guide RNA and the gene-specific genotyping primers (GF1-GR1). (B) Validation of EROS knockout (gene CYBC1) by Sanger sequencing.

• Transparent reporting form

### Data availability

The mass spectrometry proteomics data have been deposited to the ProteomeXchange Consortium via the PRIDE partner repository with the dataset identifier "PXD024659". The plasma membrane profiling mass spectrometry data has the identifier "PXD025149".

The following datasets were generated:

| Author(s) | Year | Dataset title | Dataset URL | Database and Identifier |
|---|---|---|---|---|
| Wright J, Choudhary J | 2021 | EROS/CYBC1 CONTROLS THE PHAGOCYTE RESPIRATORY BURST VIA DIRECT INTERACTION WITH GP91PHOX AND HAS A NOVEL ROLE IN NLRP3 INFLAMMASOME ACTIVATION | https://www.ebi.ac.uk/pride/archive/projects/PXD024659 | PRIDE, PXD024659 |
| Williamson J, Lehner P | 2021 | EROS is a selective chaperone regulating the phagocyte NADPH oxidase and purinergic signalling | https://www.ebi.ac.uk/pride/archive/projects/PXD025149 | PRIDE, PXD025149 |

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

# Appendix 1

**Appendix 1—key resources table**

| Reagent type (species) or resource | Designation | Source or reference | Identifiers | Additional information |
|---|---|---|---|---|
| Genetic reagent (*Mus musculus*) | C57BL6/N | Jackson Laboratory | RRID:IMSR JAX:005304 | Control mice |
| Genetic reagent (*M. musculus*) | Eros/bc017643<sup>tm1a/tm1a</sup> | PMID:28351984 | | Eros knockout |
| Genetic reagent (*M. musculus*) | Cybb<sup>tm1Din</sup> | Jackson Laboratory | RRID:IMSR JAX:002365 | gp91*phox* knockout |
| Cell line (*M. musculus*) | NIH3T3 | American Type Culture Collection | CRL-1658 | |
| Cell line (*M. musculus*) | RAW 264.7 Eros FLAG-tagged | PMID:28351984 | | Macrophages overexpressing tagged Eros |
| Cell line (*Cercopithecus aethiops*) | COS-7 | American Type Culture Collection | CRL-1651 | |
| Cell line (*Homo sapiens*) | Control and EROS-deficient human iPS-derived macrophages | This paper | | Generated through CRISP-Cas9 technology |
| Cell line (*H. sapiens*) | HEK293, HEK293T | American Type Culture Collection | CRL-1573 CRL-3216 | |
| Cell line (*H. sapiens*) | HEK293-F | Thermo Fisher | R79007 | Also known as FreeStyle 293F Cells |
| Cell line (*H. sapiens*) | HEK293, HEK293 *STT3A*<sup>-/-</sup>, HEK293 *STT3B*<sup>-/-</sup> | PMID:26864433 | | Prof Neil Bulleid (University of Glasgow) |
| Cell line (*H. sapiens*) | PLB985, EROS-deficient PLB985 (clone 14, clone 20) | DSMZ, PMID:30312704 | ACC 139 | Control and EROS knockout lines |
| Cell line (*H. sapiens*) | PLB985 EROS-GFP | PMID:30312704 | | Overexpression of EROS |
| | Anti-gp91*phox* (mouse monoclonal) | | sc-130543 | (1:2000) |
| | Anti-p22*phox* (rabbit polyclonal) | | sc-20781 | (1:1000) |
| Antibody | Anti-p22*phox* (mouse monoclonal) | Santa Cruz Biotechnology | sc-130550 | (1:500) |
| | Anti-C17ORF62/EROS (rabbit polyclonal) | | HPA045696 | (1:1000) |
| Antibody | Anti-P2X7 (rabbit polyclonal) | Atlas Antibodies | HPA044141 | (1:500) |
| | Anti-P2X1 (rabbit polyclonal) | | APR001 | (1:250) |
| | Anti-P2X4 (rabbit polyclonal) | | APR002 | (1:500) |
| Antibody | Anti-P2X7 (rabbit polyclonal) | Alomone | APR004 | (1:500) |

*Appendix 1 Continued on next page*

*Appendix 1 Continued*

| Reagent type (species) or resource | Designation | Source or reference | Identifiers | Additional information |
|---|---|---|---|---|
| | Anti-vinculin (rabbit polyclonal) | | 4650 | (1:1000) |
| Antibody | Anti-BiP (rabbit polyclonal) | Cell Signaling Technology | C50B12 | (1:1000) |
| | Anti-actin (rabbit polyclonal) | | ab8227 | (1:2000) |
| Antibody | Anti-α-tubulin (mouse polyclonal) | Abcam | ab7291 | (1:1000) |
| Antibody | Anti-STT3A (rabbit polyclonal) | ProteinTech | 12034-1 | (1:1000) |
| Antibody | Anti-calnexin (mouse monoclonal) | Invitrogen | MA3-027 | (1:1000) |
| | Anti-FLAG-M2 (mouse monoclonal) | Sigma | F3165 | (1:500) |
| Antibody | Anti-FLAG (rat monoclonal) | BioLegend | 637303 | (5 µg) |

