## [Editor Report]

This valuable study focus follows this group's previous work on EROS and NOX2. In this current study, the authors examine neutrophil EROS in the generation of superoxide by the NADPH oxidase. They convincingly demonstrate how EROS is involved in the maturation of gp91phox and expand our knowledge of the role of EROS in regulating the expression of the P2x7 ion channel. This work will be of interest to biochemists and immunologists.

---

## [Decision Letter]

**Decision letter after peer review:**

Thank you for submitting your article "EROS controls NOX2 and P2X7 destiny: a rheostat for the innate and adaptive immune system" for consideration by *eLife*. Your article has been reviewed by 3 peer reviewers, and the evaluation has been overseen by a Reviewing Editor and Satyajit Rath as the Senior Editor. The following individual involved in the review of your submission has agreed to reveal their identity: Christine Winterbourn (Reviewer #1).

Essential revisions:

Please address the concerns of the reviewers comments below.

Please make it clear what are the novel findings of the present study and their importance.

Please clarify the roles of EROS in Nox1 and Nox4.

1. In Figure 1 The lower EROS expression when gp91phox is expressed warrants a comment. Figure 1 G. Please explain what fold change represents. From F, zero time expression appears much more than the 1.5 fold higher shown in G for the EROS-expressing cells. This needs explaining. With the very high error bars (presumably for the EROS sample although this is not clear) overlapping zero I find it hard to conclude anything from this figure.

2. P 9 line 9 states that Figure 1H shows that cycloheximide increases expression. Yet it appears from the legend that cycloheximide is present in all samples and it is EROS that increases expression. Please clarify this anomaly.

3. Figure 3A&B and p12 1st para. The identification of OST as a binding partner is interesting and a significant novel finding. However, the presentation of this information appears to me to be unduly complex and more information is required. Not all the readers will be familiar with the details of SAINTexpress methodology and more explanation of what is being shown would be helpful. At the least, a supplementary Table of the 59 identified proteins would be helpful, plus information on controls to establish selective pull down by EROS and on how the blue spots in A relate to the proteins. Also please make it clearer which of the proteins in B were identified and the relevance of showing all the steps in the pathway.

4. Figure 6. This contains a large amount of information. Although interesting, I am concerned that the authors may be trying to include too much at the expense of the necessary detail for some of the experiments. For example, the EROS -/- +ATP scattergram on the left of Figure 6E does not seem to agree with the right hand graph. I would also like to see the mean values for the 5 experiments in Figure 6G shown. Most importantly, insufficient information is given for Figure 6H. I don't think I missed it but I could find no details about the experiment in the Methods section. We need to know more about exactly how many animals in were in each group (death of 1 animal appears to equate to 5% of total – how does this relate to >10 in total), how signs of illness were monitored and related to death, and generally more about the conditions of the experiment. Alternatively this may be better left to a more detailed study.

5. Details on the influenza A virus infection experiments in mice (figure 6h) are missing. Information on the used dose and virus strain for infection are important, as the outcome could be different upon high or low dose infection. It remains unclear if reduced mortality is observed due to reduced adverse inflammatory signalling or if lack of Nox2 and inflammasome activity affects the virus burden. In addition, how does the EROS knockout mouse compare to gp91phox knockout or P2X7 inhibition alone? This could be discussed in the Discussion section.

6. Provide additional controls for the possible construct misfolding and ER stress-related issues discussed above. For example, expression of other ER chaperones, better assessment of roles for endogenous Nox subunits vs. EROS.

7. Additional control experiments for the OST association in the ER are required.

8. Additional clarification of the roles of gp91phox and ROS in influenza infection vs. the loss of P2X7. E.g., compare survival of P2X7 -/- vs. EROS -/-.

9. The Y2H experiment from Figure 2G does not appear very strong. If the authors could provide a better example or clarify the relevance of this experimental result, this would strengthen the data.

10. The specificity of NGI-1 should be commented and supported with more data.

*Reviewer #1 (Recommendations for the authors):*

Please make sure all abbreviations and labels in all figures are defined.

Readability would be improved by breaking the text into more paragraphs.

P12 line 0. More straightforward to simply say …candidate interacting proteins.

Figure 3F. Please comment on why NGI-1 gives a large increase in EROS expression.

*Reviewer #2 (Recommendations for the authors):*

Details on the influenza A virus infection experiments in mice (figure 6h) are missing. Information on the used dose and virus strain for infection are important, as the outcome could be different upon high or low dose infection. It remains unclear if reduced mortality is observed due to reduced adverse inflammatory signalling or if lack of Nox2 and inflammasome activity affects the virus burden.

Also, how does the EROS knockout mouse compare to gp91phox knockout or P2X7 inhibition alone? This could be discussed in the Discussion section.

Figure 2D should state that succinyl acetone was added.

*Reviewer #3 (Recommendations for the authors):*

1. Make clear what are the novel findings of the present study and their importance.

2. Provide additional controls for the possible construct misfolding and ER stress-related issues discussed above. For example, expression of other ER chaperones, better assessment of roles for endogenous Nox subunits vs. EROS.

3. Additional control experiments for the OST association in the ER.

4. Clarify the roles of EROS in Nox1 and Nox4.

5. Additional clarification of the roles of gp91phox and ROS in influenza infection vs. the loss of P2X7. E.g., compare survival of P2X7 -/- vs. EROS -/-.

6. The Y2H experiment from Figure 2G does not appear very strong. If the authors could provide a better example or clarify the relevance of this experimental result, this would strengthen the data.

7. The specificity of NGI-1 should be commented and supported with more data.

8. Figure suppl 1F: the actin loading is unequal. Please provide quantification.

9. Figure 1H: +/- symbols for EROS appear to be mismatched.

10. Figure 5B: loading is unequal. Please provide quantification.

11. Figure 5 data do not allow affirming the mechanism of P2X7 "is analogous" to that of gp91phox.

---

## [Author Response]

Essential revisions:Please address the concerns of the reviewers comments below.Please make it clear what are the novel findings of the present study and their importance.

Our study offers several new insights compared to previous work, which we outline below. We have now also re-written the discussion around this structure in addition to making reference to the relevant literature. This should make it easier for readers of the journal to discern the new advances.

(i) We use a granular mass spectrometry detecting up to 8000 proteins in different cell types to clearly define the range of proteins that EROS can regulate in context of the entire proteome. We used both whole cell and plasma membrane profiling to show just how specific EROS is in its physiological effects. This is a significant advance on our previous mass spectrometry work. It sets the platform for advances in our understanding of the way EROS controls both NAPDH oxidase and P2X receptor biology, which we elaborate on below.

(ii) We present an EROS-deficient iPS derived human macrophage line for the first time. This recapitulates the effects seen in cell lines and patients for gp91*phox*-p22*phox* and will be a useful platform in which to interrogate the effects of EROS in the human immune system. It also demonstrates for the first time in a primary human cell that EROS deficiency leads to P2X7 deficiency. This is likely to be an important part of the human syndrome.

(iii) We show, by two different techniques that EROS binds gp91*phox* directly and that it is needed very early in gp91*phox* biogenesis, in contrast to p22*phox*. It preferentially stabilises the gp91*phox* 58kDa precursor before heme incorporation takes place. We conclusively demonstrate that the primary effect in this context is on gp91*phox* and the effect on p22*phox* is secondary. We show that it can associate with the gp91*phox*-p22*phox* heterodimer up until export from the ER. Much work has been done to characterise the latter stages of gp91*phox* maturation (Porter et al., 1996, Yu et al., 1997, Yu et al., 1999, DeLeo et al., 2000, Beaumel et al., 2014). This study thus offers unique insight into the early stages of the protein’s biosynthesis, on which all subsequent events depend. Our BiBiT split luciferase assay also describes a useful platform to model the direct interaction between EROS and gp91*phox* in future i.e. in dissecting whether point mutations in the protein might disrupt the interaction. We also pull down endogenous gp91*phox* and western blot endogenous EROS.

(iv) These early stages involve the OST-complex, which is required for the initial glycosylation events, folding and insertion into the ER membrane (Gemmer and Forster, 2020). The OST complex adds a preformed oligosaccharide to an asparagine residue on a newly synthetized protein (Cherepanova et al., 2016). Murine gp91*phox* has one asparagine residue while human gp91*phox* has three asparagine residues that are glycosylated (Harper et al., 1985, Wallach and Segal, 1997). Using PNGaseF and EndoH-treated lysates as a reference, we demonstrated that EROS immunoprecipitated with partially glycosylated gp91*phox*. Blockade of OST-mediated N-glycosylation, through chemical inhibition or use of cells lacking the catalytic subunit STT3A, impacts on the gp91*phox* glycosylation pattern upon co-transfection with EROS in HEK293 cells but does not reduce the abundance of gp91*phox* per se as does EROS deficiency. Hence, we propose that EROS works immediately post-translationally, stabilizing gp91*phox* so that the high mannose precursor can be synthetized and undergo further maturation, including heme incorporation.

(v) We show that EROS can increase the abundance of NOX1, gp91*phox* and NOX4 in co-transfection assays and that it can directly bind NOX2, NOX3 and NOX4 in yeast-2-hybrid assays. For NOX2, we utilise two different yeast two hybrid platforms. This contrasts with its inability to augment the expression of NOX5 or p22*phox* or bind them in yeast-2-hybrid. It suggests that EROS likely binds a conserved motif in NOX proteins that depend on p22*phox* for stability.

(vi) Despite the ability to bind NOX4 and increase its abundance, EROS does not control NOX4 expression. We extended our previous work to show no effect on the abundance of NOX4 in either heart or kidney in EROS knockout mice. Given our findings above, this does not rule out the possibility of EROS regulating NOX4 in **some** contexts. However, NOX4 expression is unaffected by EROS deficiency in those organs in which NOX4 is most highly expressed. The same is true of NOX1. While we cannot rule out scenarios in which EROS might regulate its expression, but it does not do so in the colon.

We also make new insights into the biology of the control of P2X7 expression by EROS, in addition to our work on human iPS-derived macrophages.

(vii) We have demonstrated immunoprecipitation of native P2X7 by EROS in both mass spectrometry and by western blot in a physiologically relevant cell type, the RAW264.7 macrophage. In addition, by performing an immunoprecipitation and mass spectrometry interactome, we have shown that this is a strong, high-confidence interaction when taking an unbiased approach and compared with all other proteins in the proteome.

(viii) We have extended the findings to show a modest effect on P2X1 but no effect on P2X4, again contextualising the interaction with P2X7.

(ix) We have shown that EROS physiologically regulates P2X7 expression. Increased EROS expression leads to increased P2X7 expression in RAW264.7 macrophages.

(x) We have shown clear effects in eight primary cell types of the mouse. Several splice variants exist in different tissues and it is important to clarify whether EROS has cell type specific effects on P2X7 expression.

(xi) A particular strength of our study is that we show marked in vivo sequelae of the lack of P2X7. EROS deficiency leads to profound susceptibility to bacterial infection but protects mice from infection with influenza A.

Please clarify the roles of EROS in Nox1 and Nox4.

We have included yeast-2-hybrid analysis of the association of EROS with NOX4, relative to NOX5 and p22*phox* and further western blot experiments of NOX1 expression in the colon and NOX4 expression in kidney and heart from EROS knockout mice in Figure 2 figure supplement 1H-I.

1. In Figure 1 The lower EROS expression when gp91phox is expressed warrants a comment. Figure 1 G. Please explain what fold change represents. From F, zero time expression appears much more than the 1.5 fold higher shown in G for the EROS-expressing cells. This needs explaining. With the very high error bars (presumably for the EROS sample although this is not clear) overlapping zero I find it hard to conclude anything from this figure.

We used transient transfection of gp91*phox* and EROS constructs for the experiments in HEK293, and NIH3T3. As such, the expression level of each construct is variable depending on the transfection efficiency. However, while the 58kDa form of gp91*phox* is reproducibly increased by EROS, EROS expression is not consistently lowered by gp91*phox* in all our experiments (see Figure 1 —figure supplement 1D for example). Moreover, absence of gp91*phox* does not impact on EROS expression in primary bone marrow derived macrophage deficient in gp91*phox* (Figure 1 —figure supplement 1F). Hence, the lower EROS expression when gp91*phox* is expressed, in some experiments, is likely due to the transient transfection than anything else.

Figure 1: The information on fold change has now been added to the figure legend. Presumably the reviewer meant Figure G left panel (not F, which is the p22*phox* western blot). The image in G (left panel) is a representative image of 4 different transfection experiments and the 1.5-fold change in EROS-expressing cells observed in G (right panel) is the mean fold change obtained from these 4 experiments. The errors bars reflect the variability of the transient transfection efficiency when expressing gp91*phox* and EROS vectors. However, the presence of EROS reproducibly reduces the rate of gp91*phox* degradation across these various experiments. We agree, the error bars are high but unfortunately it is the limitation of the approach which is why we also used the PLB985 model and lentivirus transduction system to corroborate our findings.

2. P 9 line 9 states that Figure 1H shows that cycloheximide increases expression. Yet it appears from the legend that cycloheximide is present in all samples and it is EROS that increases expression. Please clarify this anomaly.

The text in Page 9 has been amended to state the role of EROS in increasing gp91*phox* expression as seen in Figure 1H. We apologise if this was unclear.

3. Figure 3A&B and p12 1st para. The identification of OST as a binding partner is interesting and a significant novel finding. However, the presentation of this information appears to me to be unduly complex and more information is required. Not all the readers will be familiar with the details of SAINTexpress methodology and more explanation of what is being shown would be helpful. At the least, a supplementary Table of the 59 identified proteins would be helpful, plus information on controls to establish selective pull down by EROS and on how the blue spots in A relate to the proteins. Also please make it clearer which of the proteins in B were identified and the relevance of showing all the steps in the pathway.

We have now clearly stated in the Results section that we performed 4 independent biological replicate FLAG purifications from EROS-FLAG cells and 4 replicates from untagged cells as negative controls in order to control the selectivity of the EROS-FLAG pulldown. We have also included details on how the SAINTexpress algorithm (Teo et al., J Proteomics 2014, PMID: 24513533) was used to assign probabilities of EROS-specificity to the proteins identified in EROS-FLAG affinity purification-mass spectrometry experiments. SAINTexpress is a widely used algorithm in the AP-MS field that calculates a statistical model based on the abundance of proteins in bait versus control purifications to derive a confidence measure (SAINT Probability score) of a protein being specifically enriched in bait purifications. The algorithm developers suggest applying a score cut-off between 0.7-0.9. In this study we chose to be very stringent and applied a threshold of 0.9, which resulted in a False Discovery Rate (FDR) of less than 1%.

We apologise that what we displayed in Figure 3A and 3B was unclear. In Figure 3A, each blue or red spot is one of the 856 proteins identified in any one of the FLAG affinity purifications, represented by its average abundance across EROS-FLAG purifications (by proxy number of peptide spectrum matches) and its SAINT probability score (i.e., measure of EROS-specificity – the higher, the better). We chose to mark in red some of the proteins that were of particular relevance to the study, so that the reader could quickly judge visually their abundance and specificity in relation to all other identified proteins. We would like to point out that Figure 3B is not showing a pathway, although the shape of the network might induce one to believe so. In Figure 3B we have displayed all 59 proteins that passed the SP>=0.9 threshold, and the physical interactions between them that have been reported in the literature (from STRING database). We have now modified the Figure 3 legend to clarify these points. The table of the 59 identified proteins have been provided as a Source Data as per the journal request but not as a proper figure or table.

4. Figure 6. This contains a large amount of information. Although interesting, I am concerned that the authors may be trying to include too much at the expense of the necessary detail for some of the experiments. For example, the EROS -/- +ATP scattergram on the left of Figure 6E does not seem to agree with the right hand graph. I would also like to see the mean values for the 5 experiments in Figure 6G shown. Most importantly, insufficient information is given for Figure 6H. I don't think I missed it but I could find no details about the experiment in the Methods section. We need to know more about exactly how many animals in were in each group (death of 1 animal appears to equate to 5% of total – how does this relate to >10 in total), how signs of illness were monitored and related to death, and generally more about the conditions of the experiment. Alternatively this may be better left to a more detailed study.

We have included the mean values for the 5 experiments in Figure 6G as a graph. We thank the reviewer for pointing out the missing details for Figure 6H, they have been added. We have also included the details of the experiments in the “Animal use paragraph” of the Methods section.

5. Details on the influenza A virus infection experiments in mice (figure 6h) are missing. Information on the used dose and virus strain for infection are important, as the outcome could be different upon high or low dose infection. It remains unclear if reduced mortality is observed due to reduced adverse inflammatory signalling or if lack of Nox2 and inflammasome activity affects the virus burden. In addition, how does the EROS knockout mouse compare to gp91phox knockout or P2X7 inhibition alone? This could be discussed in the Discussion section.

The missing details on experiment in Figure 6H has been addressed also as per Reviewer 1 request. We thank the reviewer for this suggestion, this has been added as a new paragraph in the discussion.

6. Provide additional controls for the possible construct misfolding and ER stress-related issues discussed above. For example, expression of other ER chaperones, better assessment of roles for endogenous Nox subunits vs. EROS.

We have provided in a separate file for just the reviewer, the western blot analysis of the expression level of BiP in HEK293 cells transfected with each fluorescently tagged construct used in this study using *STT3A*^-/-^ and *STT3B*^-/-^ HEK293, as positive control (Cherepanova et al., Sci Rep 2016; PMID: 26864433) and negative control of the induction of BiP, respectively. The figure shows that there is no significant induction of ER-stress, measured by BiP, when these constructs are transfected. Indeed, the level of BiP in HEK293 cells transfected with mouse GFP-tagged gp91*phox*, pEGFP-N2-STT3A, pmRFP-N2-EROS, human mRFP-tagged gp91*phox*, mouse GFP-tagged P2X7 was comparable to the level seen in untransfected cells and *STT3B*^-/-^ cells.

7. Additional control experiments for the OST association in the ER are required.

We have added additional data (Figure 3F and Figure 3, figure supplement 1) supporting the requirement of OST catalytic subunit STT3A in the glycosylation of gp91*phox* and further clarify the role of EROS in the early steps of gp91*phox* synthesis.

8. Additional clarification of the roles of gp91phox and ROS in influenza infection vs. the loss of P2X7. E.g., compare survival of P2X7 -/- vs. EROS -/-.

Our aim in performing the influenza experiments was to demonstrate an important in vivo correlate of loss of both gp91*phox* and P2X7. Given that both are involved in the pathology of influenza A in such models, one would predict that EROS is an important regulator of both pathways in vivo*,* then EROS knockout mice should be protected and indeed, they are. This demonstrates an important aspect of EROS physiology and also highlights that EROS inhibition may be a potential therapeutic avenue in several viral pneumonia. The exact cellular mechanisms underlaying the protection of EROS knockout mice are likely a composite of inhibition of the two pathways. Given that this involves effects on several cell types in both innate and adaptive immune systems, dissecting the exact mechanisms would represent an entire separate study and is beyond the scope of the current manuscript. We have however discussed the contributions of the two pathways in our revised Discussion section.

9. The Y2H experiment from Figure 2G does not appear very strong. If the authors could provide a better example or clarify the relevance of this experimental result, this would strengthen the data.

We have now repeated the experiment through our work with Hybrigenics. These data were generated using the same protocol as that which showed the original interaction. In fact, the repeat experiment demonstrated the interaction even more clearly, thus providing a better example for Figure 2G. We also used a different yeast 2 hybrid protocol as described in Luck et al., (Nature 2020; PMID: 32296183) through a collaboration with Kerstin Spirohn and Michael Calderwood (Dana Faber Cancer Institute). This approach also demonstrated an interaction between EROS and gp91*phox* and also showed that EROS could bind NOX3 and NOX4 though not NOX5 or p22*phox*. We have therefore confirmed the direct interaction between EROS and gp91*phox* by yeast two hybrid, added important negative findings (no interaction with p22*phox* or NOX5) and new positive data (interactions with NOX4 and NOX3).

10. The specificity of NGI-1 should be commented and supported with more data.

As mentioned in our response to Reviewer 1’s comment, NGI-I inhibition of glycosylation causes the accumulation of protein that cannot be properly folded, resulting in the induction of ER-stress which we observe with the increase in stress sensor BiP in our western blot analysis of PLB985 cells treated with NGI-I and using the widely adopted stress inducer: tunicamycin as positive control (Figure 3 figure supplement 1D).

Reviewer #1 (Recommendations for the authors):Please make sure all abbreviations and labels in all figures are defined.

We have now defined all the abbreviations and labels.

Readability would be improved by breaking the text into more paragraphs.

We thank the reviewer for this suggestion. This has been done.

P12 line 0. More straightforward to simply say …candidate interacting proteins.

This paragraph has now been modified to include the additional information highlighted in point 3.

Figure 3F. Please comment on why NGI-1 gives a large increase in EROS expression.

NGI-I inhibition of glycosylation causes the accumulation of protein that cannot be properly folded, resulting in the induction of the Unfolded Protein Response (UPR) and ER-stress which manifests as an increase in the chaperone BiP (Ruiz-Canada *et al.*, Cell 2009; PMID: 19167329) and others (IRE1 α, XBP1). Thus, our result suggests that ER-stress may regulate the expression of EROS consistent with its role as an ER-chaperone necessary for the synthesis and processing of gp91*phox* and P2X7.

Reviewer #2 (Recommendations for the authors):Details on the influenza A virus infection experiments in mice (figure 6h) are missing. Information on the used dose and virus strain for infection are important, as the outcome could be different upon high or low dose infection. It remains unclear if reduced mortality is observed due to reduced adverse inflammatory signalling or if lack of Nox2 and inflammasome activity affects the virus burden.Also, how does the EROS knockout mouse compare to gp91phox knockout or P2X7 inhibition alone? This could be discussed in the Discussion section.

As stated in the Essential revision section above, we have addressed the missing details on the influenza experiments and added a new discussion paragraph:

“A particular strength of our study is that we show marked in vivo sequelae of the lack of P2X7. EROS deficiency leads to profound susceptibility to bacterial infection but protects mice from infection with influenza A. This is likely to reflect the fact that mice that are (i) deficient in gp91*phox* (ii) deficient in P2X7 (iii) treated with P2X7 inhibitors have improved outcomes following infection with influenza A and raises intriguing questions about the physiological role of EROS. Snelgrove *et al.* showed that gp91*phox* deficiency improved outcomes in influenza A. gp91*phox* knockout mice exhibited a reduced influenza titre in the lung parenchyma. Inflammatory infiltrate into the lung parenchyma was markedly reduced and lung function significantly improved (Snelgrove et al., 2006). To *et al.* then showed that the phagocyte NADPH oxidase is activated by single stranded RNA and DNA viruses in endocytic compartments. This causes endosomal hydrogen peroxide generation, which suppresses antiviral and humoral signalling networks via modification of a highly conserved cysteine residue (Cys98) on Toll-like receptor-7. In this study, targeted inhibition of endosomal reactive oxygen species production using cholestanol-conjugated gp91dsTAT (Cgp91ds-TAT) abrogates influenza A virus pathogenicity (To et al., 2017). This group went on to explore infection with a more pathogenic influenza A strain, PR8. Using the same specific inhibitor. Cgp91ds-TAT reduced airway inflammation, including neutrophil influx and alveolitis and enhanced the clearance of lung viral mRNA following PR8 infection (To et al., 2019). This group has also shown that NOX1 (Selemidis et al., 2013) and NOX4 (Hendricks et al., 2022) can drive pathogenic inflammation in influenza A, emphasising the importance of clarifying the roles of EROS in control of expression of these proteins.

In studies on P2X7, Rosli *et al.* showed that mice infected with 10^5^ PFU of influenza A HKx31 had improved outcomes if they were treated with a P2X7 inhibitor at day 3 post infection and every two days thereafter. Survival was also improved even if the inhibitor is given on day 7 post infection following a lethal dose of the mouse adapted PR8. This was associated with reduced cellular infiltration and pro-inflammatory cytokine secretion in bronchoalveolar lavage fluid, but viral titres were not measured (Rosli et al., 2019). Leyva-Grado *et al.* examined influenza A infection in P2X7 knockout mice. They infected mice with both influenza A/Puerto Rico/08/1934 virus and influenza A/Netherlands/604/2009 H1N1pdm virus. They showed that P2X7 receptor deficiency led to improved survival after infection with both viruses with less weight loss (Leyva-Grado et al., 2017). Production of proinflammatory cytokines and chemokines was impaired and there were fewer cellular hallmarks of severe infection such as infiltration of neutrophils and depletion of CD11b^+^ macrophages. It is worth noting that the P2X7 knockout strain used in this study was the Pfizer strain in which some splice variants of P2X7 are still expressed (Bartlett et al., 2014). Hence, the dual loss of the phagocyte NADPH oxidase and P2X7 in EROS knockout mice likely confers protection from IAV infection. By reducing the expression of both NOX2 and P2X7, EROS regulates two pathways that may be detrimental in influenza A and we speculate that EROS may physiologically act as a rheostat controlling certain types of immune response.”

Figure 2D should state that succinyl acetone was added.

This detail has now been added.

Reviewer #3 (Recommendations for the authors):1. Make clear what are the novel findings of the present study and their importance.2. Provide additional controls for the possible construct misfolding and ER stress-related issues discussed above. For example, expression of other ER chaperones, better assessment of roles for endogenous Nox subunits vs. EROS.3. Additional control experiments for the OST association in the ER.4. Clarify the roles of EROS in Nox1 and Nox4.5. Additional clarification of the roles of gp91phox and ROS in influenza infection vs. the loss of P2X7. E.g., compare survival of P2X7 -/- vs. EROS -/-.6. The Y2H experiment from Figure 2G does not appear very strong. If the authors could provide a better example or clarify the relevance of this experimental result, this would strengthen the data.7. The specificity of NGI-1 should be commented and supported with more data.

Point 1-7 raised by Reviewer 3 have been addressed as requested in the essential revision section above.

8. Figure suppl 1F: the actin loading is unequal. Please provide quantification.

We have elected to remove this figure. We accept that the blot is unevenly loaded. However, the result in this figure has been reproduced multiple times in the manuscript using untagged gp91*phox* and EROS constructs and in different cell lines. These are figure 1A-C, figure 1D, figure 1G, figurer 1I, figure 1 supplement 1D, figure 2 figure supplement 1D. There is no consistent effect of gp91*phox* on EROS abundance across these multiple cell lines and experimental conditions while there is always an effect of EROS on gp91*phox* abundance.

9. Figure 1H: +/- symbols for EROS appear to be mismatched.

The symbols are mismatched because the lysate for PLB985 overexpressing EROS-GFP has been loaded before the PLB985 control lysate at 0h compared to the rest of the time points where PLB985 control was loaded first.

10. Figure 5B: loading is unequal. Please provide quantification.

We take the reviewer point. However, this immunoblot is not a quantitative result but was to show that absence of gp91*phox* does not affect P2X7 expression while absence of EROS does. This later point is the focus of the rest of the Figure 5.

11. Figure 5 data do not allow affirming the mechanism of P2X7 "is analogous" to that of gp91phox.

This sentence has now been modified.